# Developmental emergence of two-stage nonlinear synaptic integration in cerebellar interneurons

**Celia Biane[1], Florian Rückerl[2], Therese Abrahamsson[2], Cécile Saint-Cloment[2], Jean Mariani[1], Ryuichi Shigemoto[3], David A DiGregorio[2], Rachel M Sherrard[1], Laurence Cathala[1,4]***

[1]Sorbonne Université et CNRS UMR 8256, Adaptation Biologique et Vieillissement, Paris, France; [2]Institut Pasteur, Université de Paris, CNRS UMR 3571, Unit of Synapse and Circuit Dynamics, Paris, France; [3]Institute of Science and Technology Austria, Klosterneuburg, Austria; [4]Paris Brain Institute, CNRS UMR 7225 - Inserm U1127 – Sorbonne Université Groupe Hospitalier Pitié Salpêtrière, Paris, France

*For correspondence:
laurence.cathala@sorbonne-universite.fr

**Abstract** Synaptic transmission, connectivity, and dendritic morphology mature in parallel during brain development and are often disrupted in neurodevelopmental disorders. Yet how these changes influence the neuronal computations necessary for normal brain function are not well understood. To identify cellular mechanisms underlying the maturation of synaptic integration in interneurons, we combined patch-clamp recordings of excitatory inputs in mouse cerebellar stellate cells (SCs), three-dimensional reconstruction of SC morphology with excitatory synapse location, and biophysical modeling. We found that postnatal maturation of postsynaptic strength was homogeneously reduced along the somatodendritic axis, but dendritic integration was always sublinear. However, dendritic branching increased without changes in synapse density, leading to a substantial gain in distal inputs. Thus, changes in synapse distribution, rather than dendrite cable properties, are the dominant mechanism underlying the maturation of neuronal computation. These mechanisms favor the emergence of a spatially compartmentalized two-stage integration model promoting location-dependent integration within dendritic subunits.

## Introduction

Dendritic integration of spatiotemporal synaptic activity is fundamental to neuronal computation, which shapes the transformation of input activity into output spiking (*Silver, 2010*). In particular, the cable properties of dendritic trees can generate isolated electrical compartments that enable nonlinear integration of local synaptic responses. These compartments increase the computational power of single neurons (*Cazé et al., 2013*; *Poirazi and Mel, 2001*) and are a prominent feature of human neurons (*Beaulieu-Laroche et al., 2018*; *Gidon et al., 2020*). Dendritic morphology and ion channel expression are developmentally regulated, but how they contribute to the maturation of neuronal computations throughout postnatal circuit formation and refinement is less well known. The observation of alterations in dendritic morphology, synaptic connectivity, density, and function in several neurodevelopmental disorders (*Marín, 2016*; *Penzes et al., 2011*) indicates that both appropriate neuronal wiring and the maturation of a neuron's integrative properties are necessary to develop fully functional neuronal networks (*Pelkey et al., 2015*).

The type and number of computations that a neuron can perform depend on the diversity of the mathematical operations used to combine synaptic inputs within dendritic trees. These can be sublinear, linear, or supralinear (*Branco and Häusser, 2011*; *Cazé et al., 2013*; *Poirazi and Mel,*

*2001*; *Tran-Van-Minh et al., 2015*; *Vervaeke et al., 2012*). Nonlinear dendritic operations depend on (1) dendritic architecture and its associated active and passive membrane properties (*Abrahamsson et al., 2012*; *Hu et al., 2010*; *Katz et al., 2009*; *Larkum et al., 2009*; *Magee, 2000*; *Magee, 1999*; *Nevian et al., 2007*; *Rall, 1967*); (2) spatial localization, density, and properties of synapses across the dendritic arbor (*Grillo et al., 2018*; *Larkum et al., 2009*; *Losonczy et al., 2008*; *Losonczy and Magee, 2006*; *Menon et al., 2013*; *Schiller et al., 2000*; *Williams and Stuart, 2002*); and (3) the spatiotemporal synaptic activity pattern (*Bloss et al., 2018*; *Grillo et al., 2018*; *McBride et al., 2008*; *Scholl et al., 2017*; *Xu et al., 2012*). All these factors change during neuronal circuit maturation through cell-autonomous or activity-dependent processes (*Sigler et al., 2017*; *Katz and Shatz, 1996*). Indeed the maturation of neuronal excitability and morphology (*Cathala et al., 2003*; *Cline, 2016*; *McCormick and Prince, 1987*; *Zhang, 2004*) is associated with restriction of neuronal connectivity to subcellular compartments (*Ango et al., 2004*), activity-dependent synaptic rearrangement (*Chen and Regehr, 2000*; *Cline, 2016*; *Kwon and Sabatini, 2011*; *Li et al., 2011*), and the maturation of excitatory (*Cathala et al., 2003*; *Hestrin, 1992*; *Koike-Tani et al., 2005*; *Lawrence and Trussell, 2000*; *Taschenberger and von Gersdorff, 2000*) and inhibitory synaptic inputs (*Ben-Ari, 2002*; *Sanes, 1993*; *Tia et al., 1996*). Despite this knowledge, how developmental changes in cellular parameters dictate dendritic operations and their associated neuronal computations, remains largely unexplored.

Interneurons are fundamental to normal circuit function throughout development. They contribute to the developmental regulation of critical periods (*Hensch et al., 1998*; *Gu et al., 2016*), are important for establishing direction selectivity in the retina (*Wei et al., 2011*), and their dysfunction is associated with neurodevelopment disorders (*Akerman and Cline, 2007*; *Le Magueresse and Monyer, 2013*; *Marín, 2016*). Cerebellar stellate cells (SCs) are parvalbumin-positive (PV+) GABAergic interneurons that share anatomical and functional features with PV+ interneurons found in the neocortex and hippocampus (*Hu et al., 2014*) such as high fidelity of synaptic transmission (*Chen and Regehr, 2000*), expression of $Ca^{2+}$-permeable α-amino-3-hydroxy-5-methyl-4-isoxazolepropionic acid (AMPA receptors) (AMPARs) (*Soler-Llavina and Sabatini, 2006*), low levels of synaptic *N*-méthyl-D-aspartate receptors (NMDARs) (*Clark and Cull-Candy, 2002*), and weakly excitable dendrites (*Abrahamsson et al., 2012*), which together provide precise temporal control of principal neuron activity (*Mittmann et al., 2005*; *Pouille and Scanziani, 2001*). SCs receive excitatory inputs from granule cells and, in turn, modulate the excitability and firing precision of the cerebellar output neurons, Purkinje cells (*Arlt and Häusser, 2020*; *Häusser and Clark, 1997*; *Mittmann et al., 2005*). The thin SC dendrites (~0.4 μm diameter) filter synaptic potentials as they propagate to the soma and confer sublinear summation of synaptic input (*Abrahamsson et al., 2012*; *Tran-Van-Minh et al., 2015*). Nevertheless, the mechanisms underlying the maturation of these dendritic operations and neuronal computation of interneurons have not been explored.

Here, we study in detail the maturation of the synaptic and integrative properties of SCs in the cerebellar cortex. We combined patch-clamp recordings with fluorescence-guided electrical stimulation, fluorescence and electron microscopy three-dimensional (3D) reconstructions, and numerical simulations, to examine synapse strength and spatial distribution. Unlike unitary inputs in other neuron types, we found that adult SCs had smaller and slower miniature excitatory postsynaptic currents (mEPSCs) than those observed in immature SCs. This could be explained by enhanced electrotonic filtering since immature SCs are thought to be electrotonically compact (*Carter and Regehr, 2002*; *Llano and Gerschenfeld, 1993*). However, we found that their dendrites are as thin as in adult SCs and capable of robust electrotonic filtering and sublinear summation of synaptic inputs. Using a novel fluorescence synaptic tagging approach, we found a significantly larger contribution of distal dendritic synapses in adult SCs, due to a substantial increase in dendritic branching combined with constant synapse density. Multicompartment biophysical modeling confirmed that developmental changes in synapse distribution could reproduce the developmental reduction and slowing of recorded mEPSCs and the increased sublinear integration observed in adult SCs. Our findings provide evidence that SCs implement different neuronal computations throughout development: a predominant global summation model in immature SCs shifts to sublinear dendritic integration in adult SCs, favoring the developmental emergence of the two-layer integration model. This work provides a mechanistic description of the maturation of neuronal computation resulting from both functional and anatomical changes in synaptic transmission and integration. Our findings and approach also provide a framework for

interpreting the functional implications of dendritic morphology and connectivity alterations on information processing within neural circuits during disease.

## Results

## AMPAR-mediated mEPSCs become smaller and slower during development

The strength and time-course of synaptic transmission are fundamental to information processing within neural networks since they influence the efficacy and temporal precision of the transformation of synaptic inputs into neuronal outputs. Excitatory synaptic inputs trigger postsynaptic conductance changes due to the opening of neurotransmitter-gated receptors, which are activated following transmitter release. These conductance changes are integrated within dendrites into local excitatory postsynaptic potentials (EPSPs) that then propagate to the cell body and contribute to somatic voltage. The strength and time-course of synaptic conductances are known to change during development (*Cathala et al., 2003*; *Chen and Regehr, 2000*; *Koike-Tani et al., 2005*) and can affect dendritic integration, which in turn may alter neuronal computation (*Tran-Van-Minh et al., 2015*).

To identify factors that shape the postnatal development of SC integrative properties, we first compared excitatory postsynaptic currents (EPSCs) recorded in acute brain slices from immature SCs soon after they reach their final position in the outer layer of the cerebellar cortex (postnatal days P13–19) and from adult SCs (postnatal days P35–57). Because of their low release probability, excitatory synapses formed by granule cell axons (parallel fibers, PFs) release on average only one synaptic vesicle per synaptic contact, despite the presence of multiple release sites per synaptic contact (*Foster et al., 2005*). We, therefore, examined these physiologically relevant 'quantal synaptic events' using somatic recordings of spontaneously occurring AMPAR-mediated miniature EPSCs (mEPSCs) in the presence of TTX to block spontaneous presynaptic activity. mEPSCs arise from the release of a single neurotransmitter vesicle and occur randomly at all synapses converging onto a single neuron. We did not examine NMDAR currents since they are located extrasynaptically and do not contribute to postsynaptic current under low-intensity and low-frequency stimulation (*Carter and Regehr, 2000*; *Clark and Cull-Candy, 2002*; *Tran-Van-Minh et al., 2016*). Therefore, AMPAR-mediated mEPSCs can provide an unbiased assessment of the effective distribution of postsynaptic strengths throughout the entire somatodendritic compartment.

We found that AMPAR-mediated mEPSCs occurred with a similar frequency at both ages (1.37 ± 0.27 vs. 1.14 ± 0.13 Hz, $p > 0.05$), but mEPSCs were significantly smaller and slower in the adult (*Figure 1A-C*). In immature SCs, the average mEPSC amplitude was 48 ± 7 pA, with 10–90% rise and decay times ($\tau_{decay}$, see Methods) of 0.16 ± 0.01 and 0.68 ± 0.06 ms, respectively. In contrast, mEPSCs from adult SCs were smaller 24 ± 2 pA ($p < 0.05$), with slower rising (0.24 ± 0.02 ms; $p < 0.05$) and decaying kinetics ($\tau_{decay}$ = 1.31 ± 0.14 ms; $p < 0.05$). The mEPSC amplitudes are consistent with those described in previous studies describing large miniature events (*Llano and Gerschenfeld, 1993*) capable of influencing immature SC firing (*Carter and Regehr, 2002*). Nevertheless, we observed that mEPSCs mature past the third postnatal week, becoming smaller and slower.

Previous studies have described developmental alterations in the glutamate content of synaptic vesicles (*Yamashita et al., 2003*) and synaptic structure (*Cathala et al., 2005*), both of which can modulate the neurotransmitter concentration in the synaptic cleft. To test whether the reduced amplitude and slower time-course could be due to alteration in *effective* amplitude and time-course of glutamate concentration ([Glut]) seen by synaptic AMPARs, we recorded mEPSCs in the presence of a rapidly dissociating, low-affinity competitive AMPAR antagonist, γDGG (*Diamond and Jahr, 1997*; *Liu et al., 1999*). Application of a submaximal concentration of γDGG (1 mM) reduced mEPSC peak amplitude (*Figure 1D*, paired $p < 0.05$) similarly at both ages (44.42% ± 4.36%, $n = 7$ in the immature vs. 42.38% ± 3.69%, $n = 9$ in the adult, $p > 0.05$; *Figure 1E*), with no apparent effect on mEPSC kinetics (*Figure 1F*, paired $p > 0.05$). This result suggests that the decreased amplitude and slowing of mEPSCs is unlikely due to a change in the synaptic [Glut]. We, therefore, explored whether postsynaptic mechanisms such as electrotonic cable filtering, as we described for adult SCs (*Abrahamsson et al., 2012*) and/or a smaller synaptic conductance (i.e., the number of activated synaptic AMPARs) could explain the changes in mEPSC during maturation.

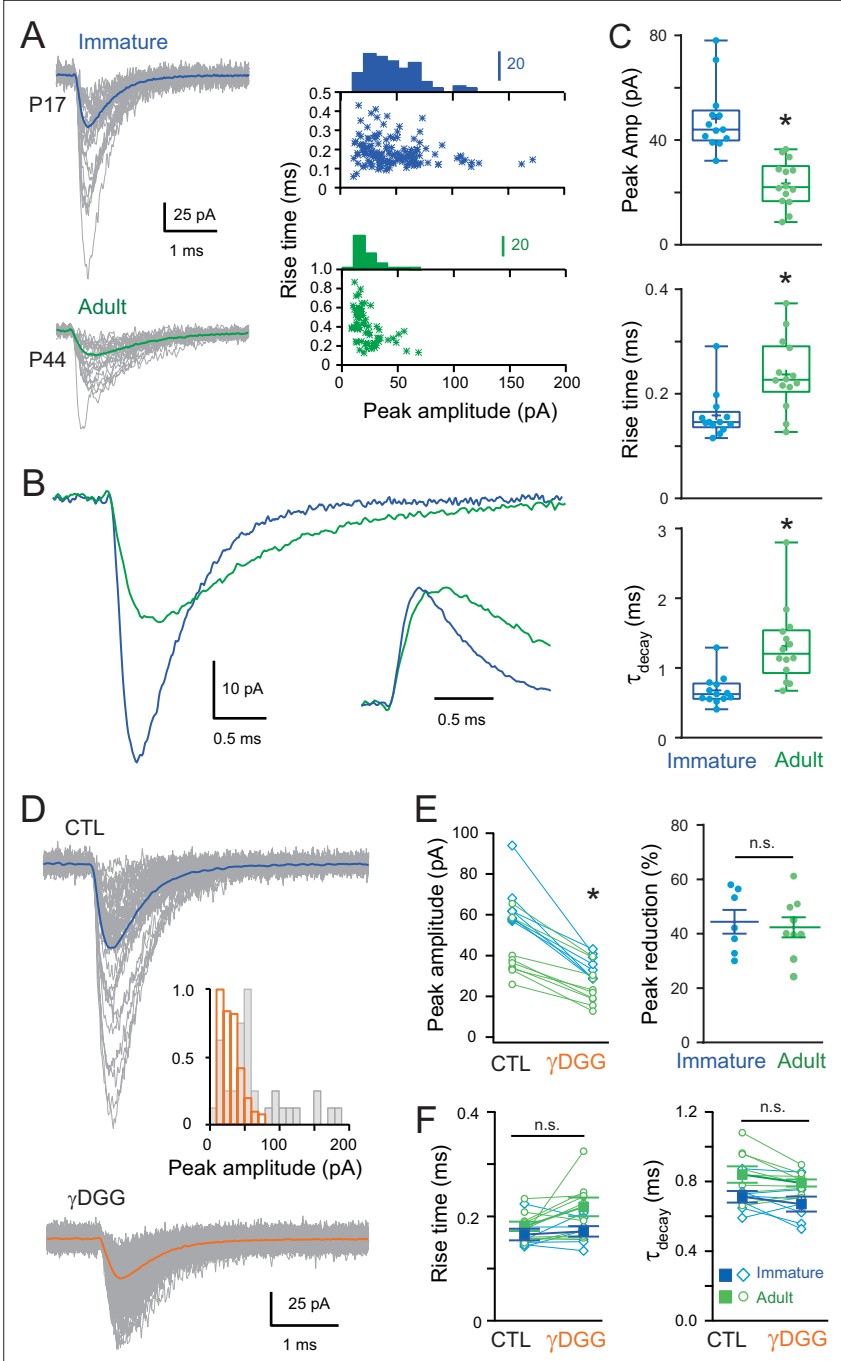

**Figure 1.** Developmental maturation of the AMPA receptor (AMPAR)-mediated miniature excitatory postsynaptic current (mEPSC) in stellate cell (SC). (**A**) Left panels show superimposed single mEPSCs (gray) and the corresponding average (bold) recorded at −70 mV obtained from a representative immature SCs (P17, blue trace) and adult SCs (P44, green trace). Right panels show plots of the 10–90% rise time versus peak amplitude, with the superimposed amplitude distributions. A significant correlation was observed in the adult SC ($p < 0.05$, Spearman rank correlation, $r = -0.59$). (**B**) Superimposed mEPSC average aligned on event onset. Inset: traces normalized to their peak. (**C**) Box and whisker plots showing the median (line) peak amplitude, 10–90% rise time and decay ($\tau_{decay}$) at both ages (P 17.4 ± 0.27 days, $n = 13$ and P 45 ± 1.75 days, $n = 14$, respectively), the 25th and 75th percentile (box), range (whiskers), and mean (+). Superimposed filled circles represent individual cells (asterisks denote $p < 0.05$; $p = 2.18e{-}5$, $p = 0.0028$, and $p = 3.0e{-}4$, respectively). (**D**) Representative examples of single mEPSC events (gray) and the corresponding average (bold) recorded from an immature SC for control (CTL) and in the presence of gamma-D-Glutamylglycine ($\gamma$DGG) (1 mM). Inset shows the corresponding mEPSC peak amplitude

*Figure 1 continued on next page*

*Figure 1 continued*

distributions. (**E**) The effect of γ DGG on mEPSCs at the two ages (*n* = 7 with p = 0.002 and *n* = 9 with p = 0.005, respectively): left panel its effect on individual mean peak amplitude for each SC (blue for immature and green for adult SC); and right panel, summary plot showing the % reduction of mEPSC peak amplitude (p = 0.83). (**F**) Plot summarizing the effect of γ DGG on mEPSC rise time (left panel; for immature SC p = 0.52 and for adult SC p = 0.09) and decay ($\tau_{decay}$, right panel, for immature SC p = 0.44 and for adult SC p = 0.53) at the two ages for individual cells (open symbols) and on population averages (± standard error of the mean (SEM)). See *Figure 1— source data 1*.

The online version of this article includes the following source data for figure 1:

**Source data 1.** Developmental maturation of the AMPAR-mediated mEPSC in SC.

## Dendritic morphology supports electrotonic filtering in both immature and adult SCs

Dendrites of adult SCs exhibit electrotonic cable filtering, which reduces the amplitude of synaptic responses and slows their time-course as they propagate to the soma (*Abrahamsson et al., 2012*), thus modifying mEPSCs recorded at the soma. We considered whether the developmental difference in mEPSC amplitude and kinetics was due to the development of electrotonic filtering. To test this hypothesis, we first estimated the dendrite diameter of immature SCs. We previously demonstrated that the small diameters (<0.5 μm) of adult SCs were responsible for slowing and reducing the amplitude of the fast AMPAR-mediated synaptic responses despite short dendritic lengths (<100 μm) (*Abrahamsson et al., 2012*). The dendritic diameter was estimated from the full-width at half-maximum (FWHM) of the fluorescence profile perpendicular to the dendrite from confocal images of live SCs aged P13–P17 filled with Alexa 488 (*Figure 2A*). Diameters ranged from 0.26 to 0.93 μm with a mean of 0.47 ± 0.01 μm (*n* = 93 dendritic segments of 18 neurons; *Figure 2B*), which is close to the average adult value of 0.41 ± 0.02 μm (range 0.24–0.9 μm, *n* = 78 dendrites; data from *Abrahamsson et al., 2012*; p < 0.05).

To understand the potential functional influence of such small diameters, we calculated the dendritic space constant (see Methods), i.e., the distance along a cable over which a steady-state membrane voltage decreases by 1/*e*. Using the estimated 0.47 μm dendritic diameter, a membrane resistance ($R_m$) of 20,000 Ω.cm² matching that measured immature SCs membrane time constant $\tau_m$ of 19 ± 2.2 ms, *n* = 16, which was similar to a $\tau_m$ of 17 ± 2.7 ms for adult SCs (*Abrahamsson et al., 2012*) and an internal resistivity $R_i$ ranging from 100 to 200 Ω.cm, we calculated the steady-state dendritic space constant (λ) to be between 343 and 485 μm, which is 3- to 5-fold longer than the actual dendritic length. The long space constants confirm that for steady-state membrane voltages, immature SCs are indeed electrically compact, as previously suggested (*Carter and Regehr, 2002*). However, the frequency-dependent space constant ($\lambda_f$; assuming that rapid mEPSCs are well approximated by a 1 kHz sine wave) was calculated to be between 46 and 60 μm (for $R_i$ of 100–200 Ω.cm, respectively). These values are similar to the dendritic lengths of immature SCs (Figure 6) and suggest that, like in adult SCs (*Abrahamsson et al., 2012*), somatic recording of EPSC originating in dendrites may be smaller and slower due to electrotonic cable filtering.

We confirmed these frequency-dependent estimations using multicompartmental biophysical models to simulate the somatic response to quantal synaptic release throughout the somatodendritic compartment. We first used an idealized SC model (*Abrahamsson et al., 2012*), and then fully reconstructed immature SC dendritic trees. For the idealized immature SC morphology (*Figure 2C*), we used an 8 μm soma diameter (8.07 ± 0.23 μm, estimated from confocal images of 31 immature SC somata) and an unbranched dendrite with a uniform 0.47 μm diameter (see mean value from *Figure 2B*), an $R_m$ of 20,000 Ω.cm² and an $R_i$ of 100–200 Ω.cm. The simulated synaptic conductance $g_{syn}$ amplitude and time-course were adjusted to reproduce recorded quantal EPSCs (qEPSC) generated by release of a single vesicle following the activation of somatic synapses (see experimental approach below and *Figure 3*). Simulated qEPSCs were large and fast for synapses located at the soma (magenta trace, *Figure 2C*), but qEPSCs evoked from synapses located on the dendrites (gray trace; at 45 μm from the soma, $R_i$ of 150 Ω.cm) were 48 % smaller and showed a 195 % slower rise time and a 180 % slower half-width. This dendritic filtering was also associated with a large increase in the local synaptic depolarization (green trace, *Figure 2C*) that would substantially reduce the local driving force during synaptic transmission onto dendrites, potentially causing a sublinear read-out of

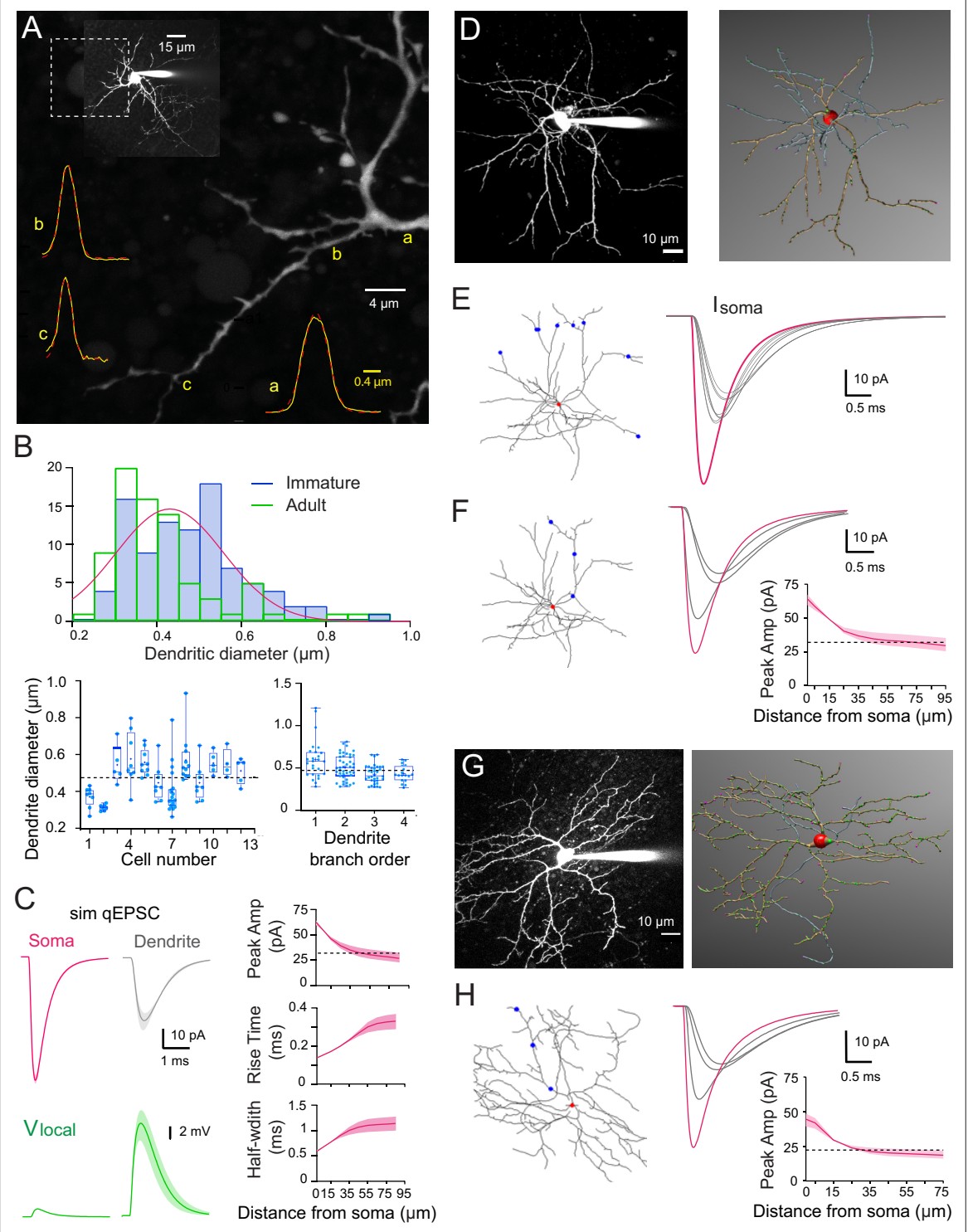

**Figure 2.** Numerical simulations of stellate cell (SC) dendrites indicate significant cable filtering and large local depolarizations in immature SC. (**A**) Maximal intensity projection of one-photon confocal images of an immature SC labeled with Alexa 488. Examples of intensity profiles (yellow line) of three dendritic locations from proximal to distal = a, (**b and c**) superimposed on the image. Dendrite diameter is approximated by the full-width half-maximum (FWHM) of the Gaussian fit of the line profile (broken red line). (**B**) Top: histogram showing the distribution of immature SC dendrite diameters from 93 dendrites (in blue) and the adult SC distribution (from *Abrahamsson et al., 2012*; in green), with a Gaussian fit indicating a mode centered at 0.43 ± 0.008 μm. Bottom left: summary box and whisker plot showing dendritic diameters for individual SCs. Superimposed filled circles represent individual dendritic branch measurements. Bottom right: summary box and whisker plot showing dendritic diameters as a function of dendritic branch order in immature SC. Filled circles represent individual dendritic diameters (with p = 0.19 between orders 1 and 2, p = 0.014 between orders 2 and 3,

*Figure 2 continued on next page*

*Figure 2 continued*

and p = 0.046 between orders 3 and 4). The dotted line indicates the mean dendritic diameter for immature SC. (**C**) Numerical simulations of somatic quantal excitatory postsynaptic currents (qEPSCs) in a passive immature SC under voltage-clamp ($C_m$ = 0.9 pF/cm$^2$, $R_m$ = 20,000 Ω.cm$^2$, and $R_i$ = 150 ± 50 Ω.cm) with a dendritic diameter set to 0.47 μm. Left: top traces show simulated qEPSCs (sim qEPSC at a $V_m$ = −70 mV) in response to a quantal synaptic conductance ($g_{syn}$) injected at the soma (magenta) and at a distance of 45 μm on a dendrite (gray trace). $g_{syn}$ was set to reproduce the experimental qEPSCs following somatic synapses activation (see *Figure 3*). Bottom traces (green), the corresponding local voltage transients at the site of synaptic conductance injection. Boundaries of shaded region indicate simulations with a $R_i$ of 100–200 Ωcm. Right: summary plot shows the distance dependence of simulated qEPSC amplitude, rise time, and half-width. Boundaries of the shaded region indicate simulations with a $R_i$ of 100–200 Ω.cm. The dotted line indicates the 50 % amplitude reduction. (**D**) Two-photon laser scanning microscopy (2PLSM) image of a P16 SC (maximal intensity projection) patch loaded with 30 μM Alexa 594 and the corresponding 3D reconstruction in *NeuronStudio* (red: soma, brown: dendrite, blue: axon). (**E**) Superimposed numerical simulation of qEPSCs in the reconstructed P16 SC (with $C_m$ = 0.9 pF/cm$^2$, $R_m$ = 20,000 Ω.cm$^2$, and $R_i$ = 150 Ω.cm) in response to a quantal conductance ($g_{syn}$) at the soma (red dot, magenta trace) or at a distance of 60 μm on six different dendritic branches (blue dots, gray traces). $g_{syn}$ was set to reproduce immature qEPSCs evoked by somatic synapses. (**F**) Simulated qEPSCs from synapse locations at the soma (red dot, magenta trace) or along a single dendrite (blue dot, gray traces). The summary plot shows the simulated qEPSC amplitudes as a function of synaptic location along the somatodendritic compartment. Boundaries of the shaded region indicate simulations with a $R_i$ of 100–200 Ω.cm. The dotted line indicates the 50 % amplitude reduction. (**G**) Same as in (**D**) but for a P42 SC. (**H**) Same as in (**F**) but with the reconstructed P42 SC and $g_{syn}$ to reproduce experimental adult somatic qEPSC. See *Figure 2—source data 1*.

The online version of this article includes the following source data for figure 2:

**Source data 1.** Numerical simulations of SC dendrites indicate significant cable filtering and large local depolarizations in immature SC.

the underlying conductance, as observed in adult SCs (*Abrahamsson et al., 2012*; *Tran-Van-Minh et al., 2016*). Examination of the amplitude, rise time, and half-width as a function of synapse distance shows that beyond ~40 μm there is little additional cable filtering (*Figure 2C*, right).

We predicted similar cable filtering with morphologically accurate passive biophysical models derived from 3D reconstructed SCs (*Figure 2D and E*). SCs were patch loaded with the fluorescence indicator Alexa 594, imaged with two-photon laser scanning microscopy (2PLSM; *Figure 2D*), and reconstructed with *NeuronStudio* (*Rodriguez et al., 2008*). The 3D reconstruction was then imported into the NEURON simulation environment, with the membrane properties indicated above. Activating a synaptic contact at 60 μm from the soma on any dendrite of the reconstructed immature P16 SC produced a simulated qEPSC that was consistently smaller and slower (gray traces) than the one produced following the activation of a somatic synapse (magenta trace; *Figure 2E*). Similarly, the activation of synaptic inputs along a dendrite at increasing distance from the soma produced soma-recorded qEPSCs that become smaller and slower with distance (*Figure 2F*), similar to those in the idealized passive model (*Figure 2C*) with a dendritic diameter matching that obtained from confocal images (*Figure 2B*). We also simulated qEPSCs from a reconstructed adult SC (*Figure 2G and H*), which showed a similar distance-dependent decrease in amplitude as for the immature SC (*Figure 2F*). These simulations suggest that, like their adult counterparts, the passive morphometric characteristics of immature SCs should also produce significant cable filtering of both the amplitude and time-course of EPSCs.

## Synaptic events are electrotonically filtered in immature SCs

To confirm modeling predictions, we next explored whether dendrite-evoked quantal events in immature SCs show evidence of cable filtering. Taking advantage of the orthogonal projection of PFs, we used parasagittal cerebellar slices to stimulate specific PF beams that are synaptically connected to well-defined regions of an Alexa 594-loaded SC by placing an extracellular electrode either above the soma or close to the distal part of an isolated dendrite branch (*Figure 3A and B*). We recorded evoked qEPSCs using whole-cell voltage clamp of the SC soma. This approach allows precise control of the location of the activated synapses, in contrast with mEPSCs that can arise from unknown synapse locations anywhere along the somatodendritic axes. Dendritic filtering could then be examined by measuring the amplitude and time-course (response width at half-peak, half-width) of these synaptic events, typically used to estimate cable filtering (*Rall, 1967*). To isolate qEPSCs, PFs were stimulated in low release probability conditions (EPSC success rate of <10 %; 0.5 mM extracellular [Ca$^{2+}$] and 5 mM [Mg$^{2+}$]). In these conditions, the average EPSC generated from all successful trials is a good approximation of the quantal current amplitude and time-course (*Silver, 2003*). When stimulating somatic synapses, qEPSCs recorded at the soma had a mean peak amplitude of 62 ± 3 pA, a 10–90% rise time of 0.14 ± 0.004 ms, and a half-width of 0.60 ± 0.02 ms (n = 25; *Figure 3C and D*). In contrast, the

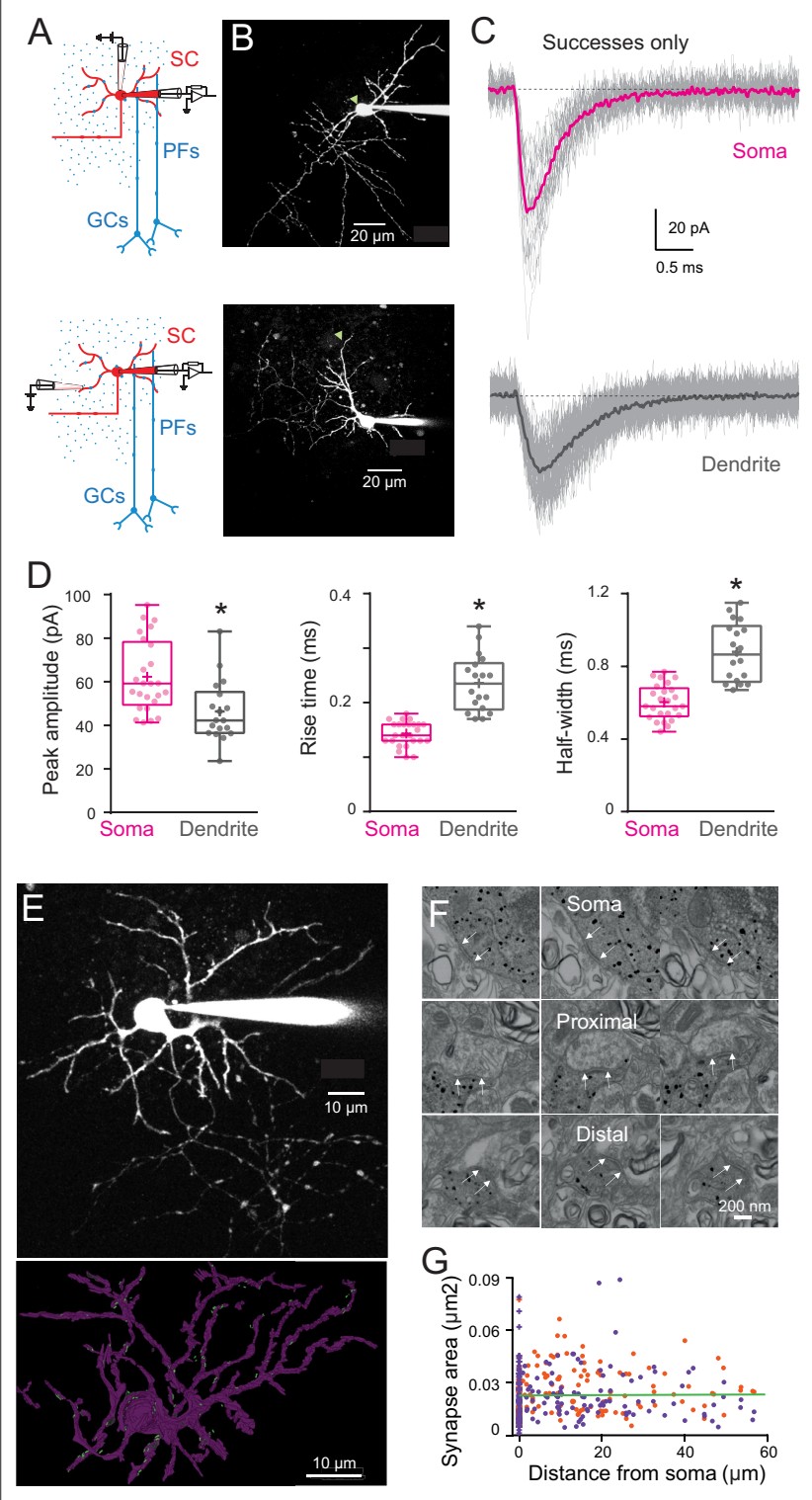

**Figure 3.** Quantal excitatory postsynaptic current (EPSC) properties and postsynaptic density (PSD) areas for synapses contacting the soma and dendrites of immature stellate cell (SC). (**A**) Diagram of a parasagittal cerebellar slice showing parallel fibers (PFs) projecting perpendicular (blue dots) to the dendritic plane of SCs (in red), allowing precise positioning of the stimulus electrode with respect to the soma (top panel) or an isolated SC dendrite (lower panel). (**B**) Two-photon laser scanning microscopy (2PLSM) images (maximal intensity projection) of two immature SC loaded with the patch-pipette with 30 μM Alexa 594 with the location of stimulating pipette indicated with a green triangle. (**C**) Single trials (gray) showing detected EPSCs (*successes only*) in response to

*Figure 3 continued on next page*

*Figure 3 continued*

extracellular stimulation at the soma (top) and on a dendrite (bottom) under low release probability conditions (external [Ca$^{2+}$]/[Mg$^{2+}$] was 0.5/5 mM; failure rate >90%). The corresponding averaged traces are in bold and represent the estimate of a quantal EPSC (qEPSC). (**D**) Box and whisker plots showing the median (line) peak amplitude, 10–90% rise time and full-width half-maximum (half-width) following somatic (magenta) or dendritic (gray) synapses activation (*n* = 25 and *n* = 18, respectively), the 25th and 75th percentile (box), range (whiskers), and mean (+). Superimposed filled circle represents individual cells (asterisks denote p < 0.05; p = 0.001, p = 4.31e−8, and p = 1.49e−6, respectively). (**E**) 2PLSM image of an Alexa 594-loaded P14 SC (maximal intensity projection) before fixation and (below) its 3D rendering after an electron microscopy (EM) reconstruction. Light dots indicate postsynaptic density (PSD) locations. Scale bar, 10 µm (**F**) Electron micrographs of an immunogold labeled SC soma with proximal and distal dendritic segments. The outer bound of excitatory synapses are indicated by arrows. Scale bar, 200 nm. (**G**) Plot of synapse area versus distance from soma. Orange and purple circles indicate data obtained from two immature SCs (P14, *n* = 172 synapses and P17, *n* = 220 synapses – total *n* = 392). The green line is a linear fit through all the points (*R*$^2$ = 0.001, p = 0.85). See *Figure 3—source data 1*.

The online version of this article includes the following source data for figure 3:

**Source data 1.** qEPSC properties and PSD areas for synapses contacting the soma and dendrites of immature SC.

---

stimulation of distal synapses, in the outer third of the dendritic field, produced somatically recorded qEPSCs that were significantly smaller (mean amplitude: 46 ± 3 pA) and slower (10–90% rise time of 0.24 ± 0.012 ms and a half-width of 0.88 ± 0.04 ms; *n* = 18; all p < 0.05, *Figure 3C and D*). Simulated qEPSCs generated from a dendritic location 45 µm from the soma (*Figure 2C*) exhibit amplitude and kinetics values with the range of experimental values. Taken together, these results are consistent with cable filtering of EPSCs as they propagate along dendrites in immature SCs.

The decreased amplitude of qEPSCs evoked in the dendrite could also be due to lower AMPAR content of dendritic synapses. As AMPAR density in SC synapses is constant (*Masugi-Tokita et al., 2007*), we used postsynaptic density (PSD) size as a proxy for the number of AMPARs. We measured PSD area of somatic and dendritic synapses from 3D electron microscopy reconstruction of immature SCs. We reconstructed the soma and the dendritic tree of two SCs (P14 and P17) loaded with Alexa 594 and biocytin (*Figure 3E*). Immunogold labeling of biocytin made it possible to identify PSDs along soma and dendrites originating from the labeled SC (*Figure 3F*). PSD area was constant along soma-todendritic axes (*Figure 3G*), ruling out synaptic scaling as a mechanism for reducing dendrite-evoked qEPSCs. Thus, the difference in qEPSC amplitude and time-course observed between somatic and dendritic synapses in immature SCs is likely due to cable filtering.

## Developmental changes in synaptic conductance amplitude, but not time-course

Because cable filtering of synaptic responses is present in both immature and adult SCs, we next explored whether differences in mEPSC amplitude between the two ages were due to the maturation of quantal synaptic conductance. To avoid the effects of cable filtering, we measured qEPSCs only at somatic synapses. In immature SCs, somatic qEPSCs were 41 % larger than those in adult SCs (62.3 ± 3.3 pA, *n* = 25 vs. 44 ± 2.4 pA, *n* = 12, *Abrahamsson et al., 2012*), p < 0.05, *Figure 4A and B*, but with no difference in the half-width (0.60 ± 0.02 ms, *n* = 25 vs. 0.59 ± 0.06 ms, *n* = 12, p > 0.05, *Figure 4B*) suggesting a reduction in postsynaptic strength but no change in kinetics.

This reduction in synaptic conductance could be due to a reduction in the number of synaptic AMPARs activated and/or a developmental change in AMPAR subunits. SC synaptic AMPARs are composed of GluA2 and GluA3 subunits associated with TARP γ2 and γ7 (*Bats et al., 2012*; *Liu and Cull-Candy, 2000*; *Soto et al., 2007*; *Yamazaki et al., 2015*). During development, GluR2 subunits are inserted to the synaptic AMPAR in an activity-dependent manner (*Liu and Cull-Candy, 2002*), affecting receptors calcium permeability (*Liu and Cull-Candy, 2000*). However, those developmental changes have little impact on AMPAR conductance (*Soto et al., 2007*), nor do they appear to affect EPSC kinetics (*Liu and Cull-Candy, 2002*); the latter being consistent with our findings. Therefore, the developmental reduction in postsynaptic strength most likely results from fewer AMPARs activated by the release of glutamate from the fusion of a single vesicle.

To confirm the reduction in synaptic AMPARs during maturation, we compared somatic PSD areas between immature and adult SCs. In immature SCs, mean somatic PSD size was 0.039 ± 0.002 µm$^2$ (*n*

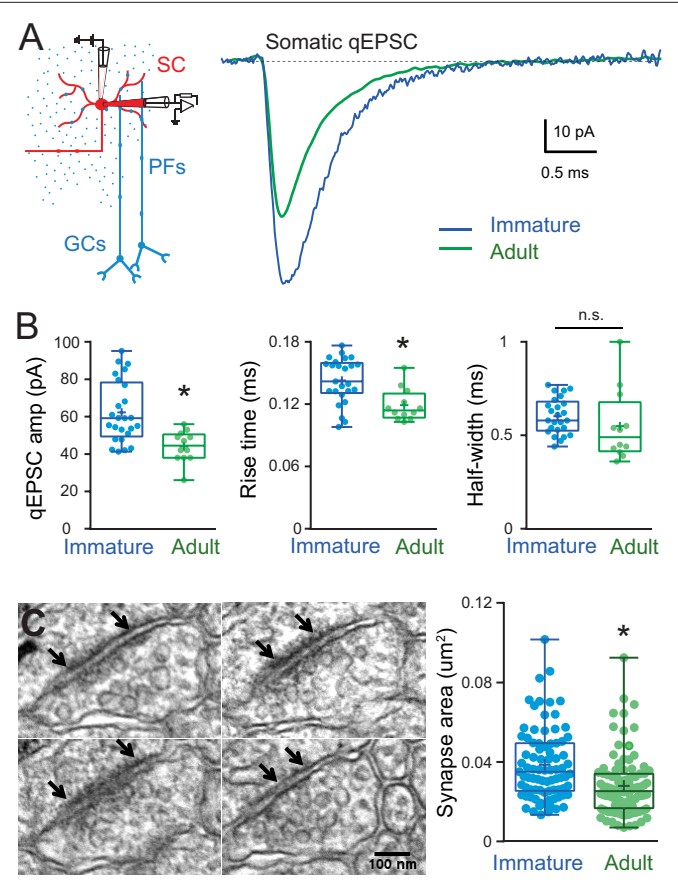

**Figure 4.** Developmental changes in somatic quantal excitatory postsynaptic current (qEPSC) properties and somatic postsynaptic density (PSD) size. (**A**) The left panel shows a diagram of a parasagittal cerebellar slice showing parallel fibers (PFs) projecting perpendicular (blue dots) to the dendritic plane of a stellate cell (SC) (in red), showing the stimulus electrode (top) position above the soma. The right panel shows the superimposition of representative qEPSC averages aligned on event onset, from immature SC (blue trace) and adult SC (green trace, from *Abrahamsson et al., 2012*). (**B**) Summary box and whisker plot showing the qEPSC amplitude, 10–90% rise time and half-width for immature (blue, *n* = 25) and adult (green, *n* = 12) SC (asterisks denote p < 0.05; p = 6.2e−4, p = 0.0035, and p = 0.074, respectively). (**C**) Serial electron micrographs of an asymmetrical synapse made by axon terminals on an immature SC soma. Right panel: summary box and whisker plot showing the synapse area obtained from immature (*n* = 83 synapses from three cells) and adult SC (*n* = 97 synapses from two cells) somata. Superimposed filled circle represent individual synapses (asterisks denote p < 0.05; p = 1.03e−5). The outer boundaries of an excitatory synapse are indicated by arrows. Scale bar is 100 nm. See *Figure 4—source data 1*.

The online version of this article includes the following source data for figure 4:

**Source data 1.** Developmental changes in somatic qEPSC properties and somatic PSD size.

= 83 synapses; *Figure 4C*), which is 39 % larger than that of adult SCs (0.028 ± 0.0015 μm², *n* = 97, p < 0.05; data obtained by the same method from *Abrahamsson et al., 2012*). The developmental reduction in PSD size is similar to the amplitude reduction of recorded somatic qEPSC and thus will contribute to the developmental reduction in mEPSC. This is supported experimentally by a slower 10–90% rise time in immature SCs compared to adult (0.14 ± 0.02 ms, *n* = 25 vs. 0.12 ± 0.004, *n* = 12, p < 0.05; *Figure 4B*, middle panel), consistent with larger immature synapses (*Cathala et al., 2005*). However, fewer synaptic AMPARs cannot explain the observed change in mEPSC time-course.

## Dendritic distribution of excitatory synaptic inputs changes during maturation

In addition to cable filtering of synaptic currents, a neuron's somatic response is also determined by the pattern of actived synapses within its dendritic tree (*Branco and Häusser, 2011*; *Tran-Van-Minh*

*et al., 2015*). Moreover, the distribution of synaptic contacts within the dendritic tree may not be uniform, as demonstrated for starburst amacrine interneurons (*Vlasits et al., 2016*) and CA1 pyramidal neurons (*Katz et al., 2009*; *Magee and Cook, 2000*). We hypothesized that changes in synapse distribution could underlie the slowing of mean mEPSCs time-course observed adult SCs. In support of this hypothesis, immature SC mEPSC rise and decay kinetics are similar to those of somatic qEPSCs (compare *Figure 1* vs. *Figure 4*), whereas adult SC mEPSC kinetics are closer to those of dendritic qEPSCs (*Figure 1* vs. *Figure 3*). Specifically, in immature SCs the mean mEPSC decay ($\tau_{decay}$ = 0.68 ± 0.06 ms) is similar to the qEPSC decay from somatic synapses (half-width = 0.60 ± 0.02 ms; p > 0.05), but significantly different from those of dendritic synapses (half-width = 0.88 ± 0.04 ms; p < 0.05). This suggests that synaptic responses from distal synapses do not participate significantly to the mean mEPSC at this developmental stage. In contrast, the adult mEPSC decay ($\tau_{decay}$ = 1.31 ± 0.14 ms) is close to the dendritic qEPSC decay (half-width = 1.14 ± 1.92, p > 0.05), but significantly different from that of somatic synapses (half-width = 0.59 ± 0.06, p < 0.05), estimated previously (*Abrahamsson*

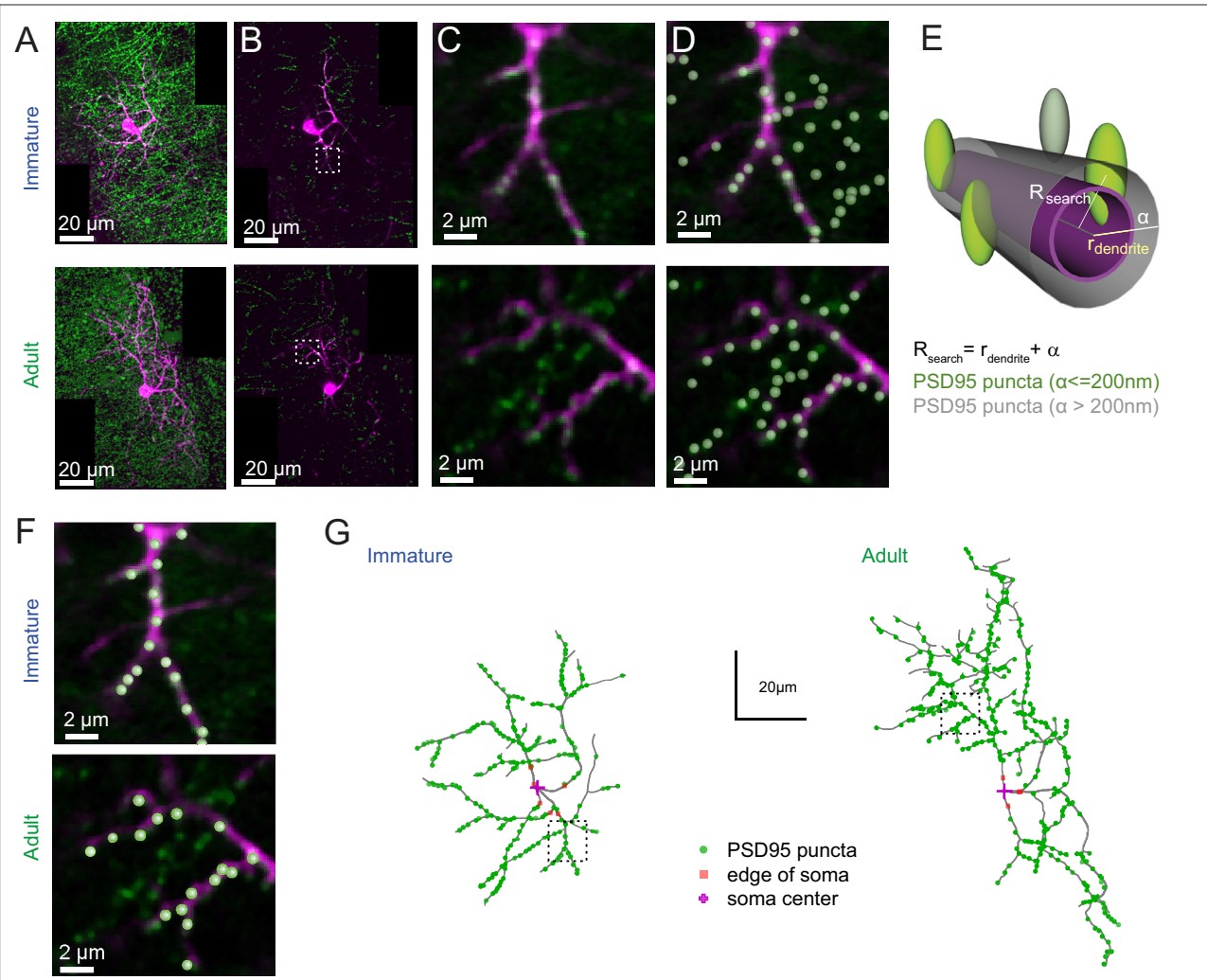

**Figure 5.** PSD95 puncta distribution on immature and adult stellate cell (SC). (**A**) Maximum intensity projection of merged images showing a P14 (top) and P42 (bottom) SC labeled with Alexa 594 (magenta) and Venus-tagged PSD95 puncta (green). (**B**) Example of single optical sections from (**A**). (**C**) Inset indicated in (**B**) showing details of a dendritic branch with Venus-tagged PSD95 puncta. (**D**) Detected PSD95 puncta on the same focal plane overlayed on the fluorescent image. (**E**) Diagram describing the criteria for assignment of PSD95 puncta to a dendrite branch. Puncta were considered as associated with the dendrite if their centers were located within a search radius ($R_{search}$) defined as the local dendritic radius $r_{dendrite}$ + α, where α = 0.2 μm. (**F**) Examples of detected PSD95 puncta connected to dendritic structure in an immature (top) and adult (bottom) SC. (**G**) Skeleton representation of the dendritic tree of an immature (left) and adult (right) SC with detected PSD95 puncta in green. The edge and the center of the soma are indicated by orange squares and magenta crosses, respectively.

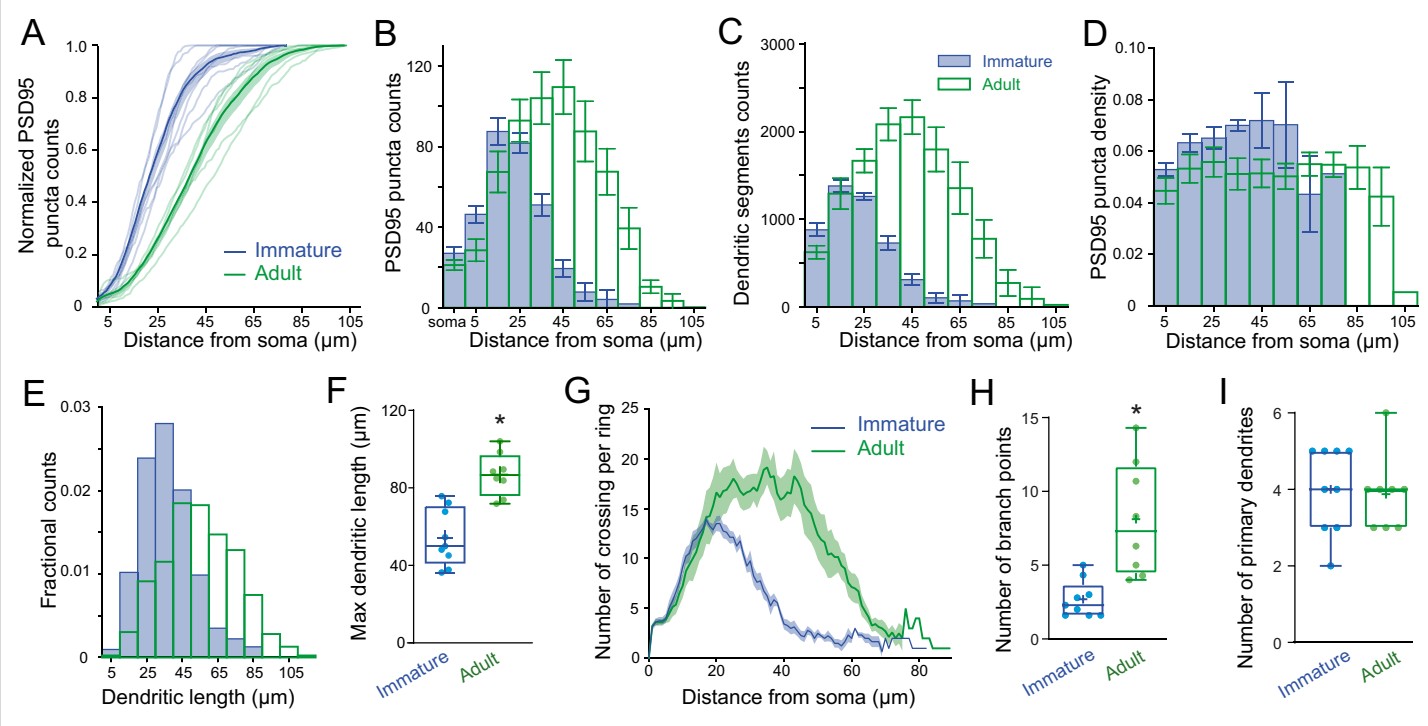

**Figure 6.** PSD95 puncta distribution and morphological analysis of immature and adult stellate cell (SC). (**A**) Cumulative plot showing Venus-tagged PSD95 puncta distributions for nine immature (blue) and eight adult (green) SCs. The bold trace is the population average (shaded region indicates SEM), showing a significant difference between immature and adult distributions (Kolmogorov–Smirnov test p < 0.0001). (**B**) Superimposed histograms of the mean PSD95 puncta count on the soma and dendrites of immature (blue) and adult (green) SC with a 10 μm increment (Pearson's median skewness: 0.38 vs. −0.1). (**C**) Superimposed histograms of the average dendritic segment count (segment length of 100 nm) for immature (blue) and adult (green) SCs (Pearson's median skewness: 0.32 vs. −0.08). (**D**) Dendritic mean PSD95 puncta density as a function of distance estimated from B and C (for all p > 0.05 except at 5 μm; MW test). (**E**) Superimposed histograms of the dendritic segment length (measured as the distance from the tip of dendrites to the soma) for immature (n = 314 dendrites, blue) and adult (n = 854 dendrites, green) SCs. (**F**) Summary box and whisker plot showing the maximal dendritic length per neuron for immature and adult SCs. Filled circles represent individual cells (asterisks denote p < 0.05; p = 6.0e−4). (**G**) Sholl analysis showing increased arbor complexity with development. (**H**) Summary box and whisker plot showing the number of branch points per primary dendrites for immature and adult SCs. Branches were defined as dendritic branches from primary dendrites that were longer than 10 μm. Superimposed filled circle represents individual cells (p = 0.0011). (**I**) Summary box and whisker plot showing the number of primary dendrites arising from the soma of immature and adult SCs, with filled circles representing individual cells (p = 0.6833). See **Figure 6—source data 1**.

The online version of this article includes the following source data for figure 6:

**Source data 1.** PSD95 puncta distribution and morphological analysis of immature and adult SC.

*et al., 2012*). These results are consistent with mEPSCs in adult SCs arising more often from more distal synapses.

To test this hypothesis, we examined the distribution of excitatory synaptic inputs along the somatodendritic compartment in immature and adult SCs. Since SC dendrites lack spines, we used transgenic mice conditionally expressing Venus-tagged PSD95 to label putative excitatory synapses. We then mapped Venus-tagged PSD95 puncta associated with the somata and dendritic trees of Alexa 594-filled immature and adult SCs (*Figure 5A–D*), defining puncta located within 200 nm of the dendritic surface (*Figure 5E*) as excitatory synapses targeting the dendrite (*Figure 5F*; see Methods). Venus-tagged PSD95 puncta within the soma and dendritic tree of nine immature and eight adult 3D-reconstructed SCs (*Figure 5G*), showed ~80 % more puncta on adult SCs (582 ± 48 puncta vs. 324 ± 21 in immature SCs, p < 0.05; of which 27 ± 3.2 and 21.17 ± 2.5 are at the soma of immature and adult SCs, respectively). Synapse distribution was assessed by counting the number of PSD95 puncta within 10 μm segments at increasing distance from the soma. In immature SCs, ~ 80 % are within 35 μm of the soma, in contrast to only ~40 % this close to the soma in adult SCs (*Figure 6A and B*). However, the ratio of detected puncta (*Figure 6B*) to dendritic segments (*Figure 6C*) shows that puncta density remained constant across the dendritic tree at both ages (*Figure 6D*). Thus, the larger

number of puncta located further from the soma in adult SCs is not due to increased puncta density with distance, but larger dendritic lengths (*Figure 6E and F*) and many more distal dendritic branches (*Figure 6G*, Sholl analysis) due to a larger number of branch points (*Figure 6H*), but not a "greater number of primary dendrites (*Figure 6I*). The similarity between the shapes of synapse (*Figure 6B*) and dendritic segment (*Figure 6C*) distributions was captured by a similarity in their skewness (0.38 vs. 0.32 for both distributions in immature and −0.10 and −0.08 for adult distributions). These data demonstrate that increased dendritic complexity during SC maturation is responsible for a prominent shift toward distal synapses in adult SCs. Therefore, if mEPSCs were generated from a homogeneous probability of release across all synapses, the bias toward distal synapses in adult SC will generate quantal responses that experience stronger cable filtering.

## Change in synapse distribution underlies developmental slowing of mEPSC

We next examined whether differences in synapse distribution could account quantitatively for the observed changes in mEPSC amplitude and time-course. We performed numerical simulations using reconstructed immature and adult SCs (*Figure 2D and G*) and a quantal synaptic conductance ($g_{syn}$) that reproduced measured immature and adult qEPSCs induced at somatic synapses (*Figure 4*). We simulated qEPSCs evoked by synaptic activation at the soma and at 10 µm intervals along the soma-todendritic axes (*Figure 7A*). Assuming that mEPSCs are generated randomly with an equal probability at all synapses, we generated a simulated *mean mEPSC* by summing the qEPSCs generated at each distance ($qEPSC_d$) each weighted by its relative frequency according to the synapse distribution (*Figure 7B*, right panel). For each dendritic segment, the obtained weighted $qEPSC_d$ describes its relative contribution to the mean mEPSC waveform (*Figure 7B*, left panel). One can see that $qEPSC_d$s arising from distal locations in the adult were relatively larger in amplitude as compared to immature SCs and therefore would contributed more to the simulated mean mEPSC waveform. As an example, the contribution of the weighted $qEPSC_d$ at 45 µm to mEPSC was small in immature SC while it was more prominent in adult SC (compare the green traces in *Figure 7B*). As a result, the mean mEPSC was smaller in adult SCs than in immature SCs (26.0 ± 0.6 pA, $n$ = 9 dendrites vs. 52.2 ± 0.4 pA, $n$ = 7 for $R_i$ 150 Ω.cm) and its time-course had slower rise (10–90% rise time = 0.24 ± 0.05 vs. 0.17 ± 0.01 ms) and decay (half-width = 1.11 ± 0.04 vs. 0.77 ± 0.01 ms; for all p < 0.05). Thus, distally generated $qEPSC_d$s dominate the mean mEPSC more strongly in adult than in immature SCs, making the mean mEPSC smaller and slower in adult SCs. Moreover, the simulated *mean mEPSCs* lay within one standard deviation of measured experimental mEPSC values (*Figure 7C and D*). Thus, by implementing the experimentally observed synapse distributions in our simulations, we could reproduce the experimental mean mEPSCs. These results demonstrate that developmental increases in dendritic branching complexity, provided that synapse density is homogeneous, can account for the changes in mEPSC kinetics (*Figure 1*).

## Influence of synapse distribution on dendritic integration of multiple synaptic inputs

We previously showed in adult SCs that the activation of dendritic synapses conveys sublinear integration as compared to the soma (*Abrahamsson et al., 2012*) due to the large local input resistance of thin dendrites resulting in synaptic depolarizations that reduced synaptic current driving force (*Bloomfield et al., 1987*; *Rall, 1967*). This also produced a distance-dependent decrease in short-term plasticity (STP). We examined whether dendritic integration in immature SCs is also sublinear by comparing STP following dendritic and somatic synapse activation. PFs targeting SC somata and dendrites were stimulated using a pair of extracellular voltage pulses with an interval of 20 ms. The paired-pulse ratio (PPR; the ratio of the amplitudes of the second vs. the first EPSC) was 2.1 ± 0.1 for somatic synapses ($n$ = 10) and decreased to 1.8 ± 0.1 for distal dendritic synapses ($n$ = 15; p < 0.05; *Figure 8A*), consistent with sublinear integration. These results were reproduced by numerical simulations of evoked EPSCs (*Figure 8B*) or EPSPs (data not shown) in the idealized passive SC model (with a synaptic $g_{syn}$ matching the recorded EPSC evoked at the soma and a 2.25 conductance ratio). These findings show that immature SC dendrites also display an STP gradient, suggesting that they are capable of sublinear integration.

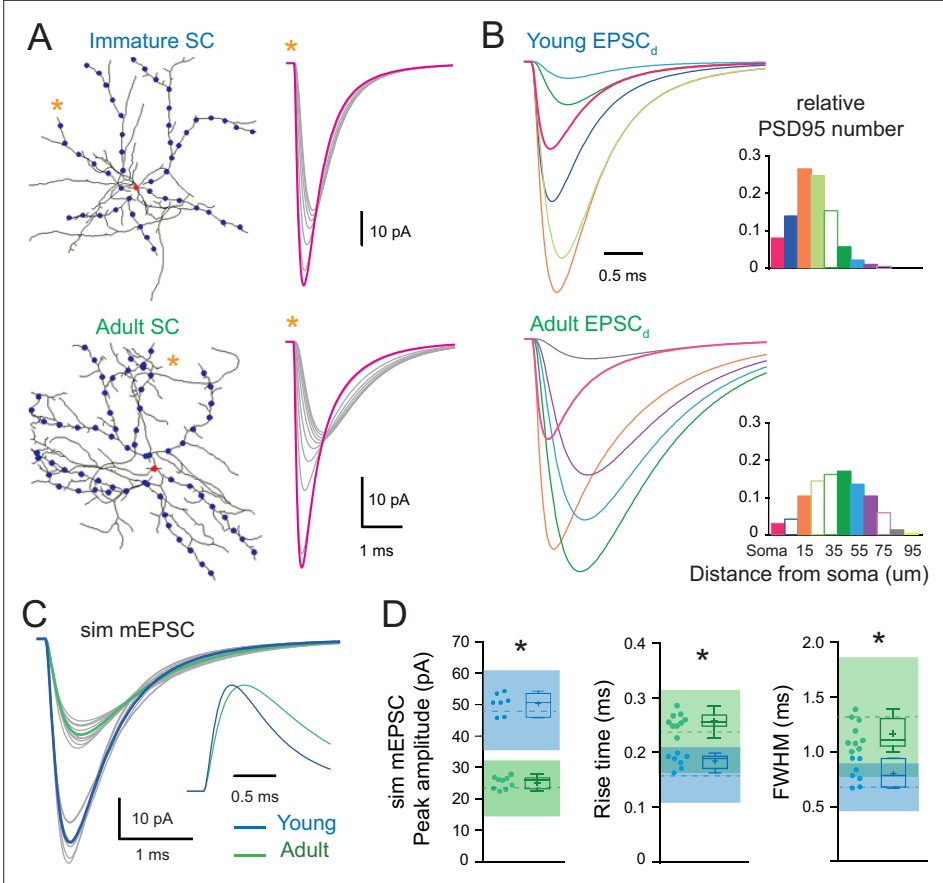

**Figure 7.** The developmental change in synaptic distribution recapitulates the developmental change in miniature excitatory postsynaptic current (mEPSC) properties. (**A**) Numerical simulations of somatic and dendritic quantal excitatory postsynaptic currents (qEPSCs) for synapses placed on the soma (red dot) and at 10 μm intervals (blue dots) along seven of the longest dendrites of a reconstructed immature (P16; top) and nine of the longest dendrites of a reconstructed adult (P42; bottom) stellate cell (SC) (with $C_m = 0.9$ pF/cm², $R_m = 20,000$ Ω.cm², and $R_i = 150 \pm 50$ Ωcm). The right panel shows an example of qEPSC for somatic (magenta) and dendritic (gray traces) synapses along a single dendrite labeled with an asterisk. The synaptic quantal conductances $g_{syn}$ were set to reproduce the experimental qEPSCs when stimulating somatic synapses at both ages. (**B**) Shows simulated qEPSCs elicited at different distances from the soma that are weighted by the relative frequency of the synapse distances (qEPSC$_d$s) extracted from the histogram to the right (derived from **Figure 6B**). For clarity, only a subset of normalized qEPSC$_d$s are displayed with a different color for somatic and each 10 μm dendritic segments and correspond to solid bars in the histogram. (**C**) Superimposed mEPSC waveforms obtained from the weighted average of qEPSC for each dendrite (gray) and the corresponding averaged simulated mEPSC (bold) for the immature (P17, blue trace) and adult (P44, green trace) SC reconstructions. Inset: traces normalized to their peak. (**D**) Summary box and whisker plots showing the median (line) peak amplitude, 10–90% rise time and decay, the 25th and 75th percentile (box), range (whiskers), and mean (+) of the sim mEPSC for the immature (blue) and adult (green) SCs. Individual dendritic mEPSCs are illustrated with filled circles (asterisks denote p < 0.05; with p = 0.0002, p < 0.0001, and p = 0.0002, respectively). The dotted line shows the experimental mEPSC average values ±1 SD (shaded region). See **Figure 7—source data 1**.

The online version of this article includes the following source data for figure 7:

**Source data 1.** The developmental change in synaptic distribution recapitulates the developmental change in mEPSC properties.

To address this possibility, we used biophysical modeling of subthreshold synaptic input–output relationships (I/O) that has been shown to accurately reproduce experimental sublinear I/Os recorded after neurotransmitter photouncaging (**Abrahamsson et al., 2012**). Evoked EPSPs were simulated (sim eEPSP) in response to increasing synaptic conductance ($g_{syn}$), equivalent to one to 20 quanta in order to encompass sparse and clustered activation of PFs (**Wilms and Häusser, 2015**) at the

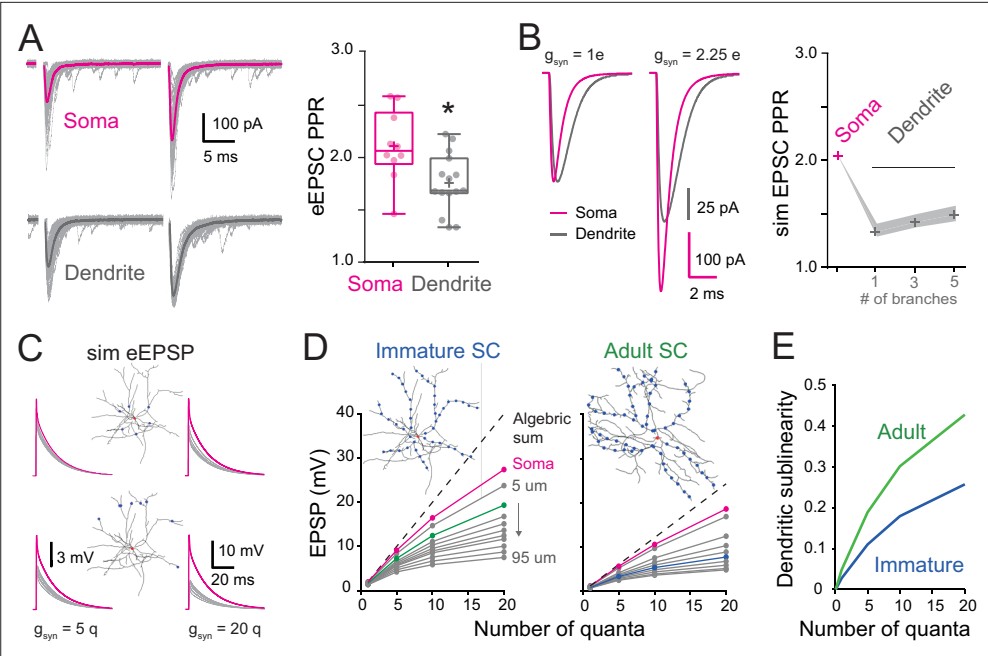

**Figure 8.** Location dependence of short-term plasticity and sublinear behavior in immature stellate cell (SC). (**A**) Single-trial excitatory postsynaptic current responses (eEPSCs, gray) and their corresponding average (bold) evoked by a pair of electrical stimuli (at 50 Hz).The stimulation electrode was placed above the soma (magenta) or on the distal portion of an isolated dendrite (gray). Right: summary box and whisker plot of paired-pulse ratio of eEPSC amplitudes (paired-pulse ratio [PPR] = eEPSC$_2$/eEPSC$_1$) for somatic synapses ($n$ = 10) and dendritic synapses ($n$ = 15). The PPR ratio was assessed from recordings where the first EPSC had an amplitude inferior to 400 pA and had a failure rate below 30 % (for somatic synapses, Amp = 239.4 ± 32 pA with a half-width = 0.88 ± 0.07 ms, $n$ = 10). Superimposed filled circles represent individual cells (asterisks denote $p < 0.05$; $p$ = 1.82e−2). (**B**) Numerical simulations of paired-pulse facilitation of eEPSCs using the idealized SC model (***Figure 2C***). Simulated eEPSCs for somatic (magenta) and dendritic (gray) synapses are superimposed, showing a lower dendrite PPR. To simulate eEPSC1, the $g_{syn}$ from ***Figure 2C*** was scaled to match the mean recorded eEPSC amplitude. The simulation of eEPSC2 was further scaled by a factor of 2.25. Right: summary plot showing the simulated PPR at the soma (magenta cross) and the lack of influence of a variable number of dendritic branches (gray crosses). Shaded region indicates simulations with a $R_i$ of 100–200 Ωcm. (**C**) Simulated evoked EPSP (sim eEPSP) under current-clamp conditions for synapses at the soma (magenta) and at 20 or 60 μm on dendrites (gray) of a reconstructed immature SC. The $g_{syn}$ peak amplitude was set to simulate 5 or 20 quanta. (**D**) Subthreshold input–output relationship obtained by plotting the average peak sim eEPSP amplitude for an increasing number of quanta versus the algebraic sum of the sim eEPSPs for a reconstructed immature (left; from ***Figure 7A***) and adult (right; from ***Figure 7A***) SC. The dotted black line has a slope of 1. The circles indicate sim eEPSP resulting from a $g_{syn}$ peak amplitude of 1, 5, 10, and 20 quanta. (**E**) Summary plot showing dendritic sublinearity (1 − (dendritic EPSP amp/soma EPSP amp) with EPSP converted to number of quanta) as a function of the number of quanta for reconstructed immature (blue) and adult (green) SCs. See ***Figure 8—source data 1***.

The online version of this article includes the following source data for figure 8:

**Source data 1.** Location dependence of short-term plasticity and sublinear behavior in immature SC.

soma and at 10 μm intervals along the reconstructed dendrites of the immature SC (***Figure 8C***). I/O plots showed that sim eEPSPs in the soma and dendrites were less than the linear sum of eEPSPs (dashed line). This sublinear summation was apparent for dendritic eEPSPs generated from synaptic conductances equivalent to one quantum for dendritic synapses and became more pronounced for distal synapses (***Figure 8D***). The sublinearity increased with increasing number of simultaneously activated quanta. These simulations show that immature dendrites demonstrate sublinear summation, supporting experimental difference between somatic and dendritic STP (***Figure 8A***).

To estimate the impact of development changes in synapse distribution on the maturation of SC computations, we compared simulated subthreshold I/Os between immature and adult SCs with their respective age-dependant $g_{syn}$ (***Figure 8D***). For immature SCs we examined subthreshold I/

Os when activating synapses at 15 μm along reconstructed dendrites (*Figure 8D*, green), a distance with the highest relative number of synaptic contacts (*Figure 7B*), and compared to I/Os generated from synapse activation at 45 μm in adult SCs (distance with the largest relative number of synapses). Sublinearity was quantified by normalizing sim eEPSPs to an sim eEPSP evoked by injecting a $g_{syn}$ of 0.1 quanta, a conductance to which the voltage is linearly related. Because large $g_{syn}$ can generate sublinear integration at the soma (*Figure 8D*), we estimated the dendrite-specific sublinearity for each $g_{syn}$ by taking the ratio between the relative sim eEPSPs amplitude (normalized to quanta) at a given distance and the normalized sim eEPSP amplitude for somatic synapses. The final estimate of dendritic sublinearity was then defined as one minus this ratio (*Figure 8E*). While both immature and adult SCs exhibited sublinear integration, dendritic sublinearity was larger in adult SCs for all postsynaptic strengths, supporting an increased difference between the two layers of integration (i.e., soma vs. dendrite). The smaller difference in sublinearity between soma and dendrite in the immature SC resulted from both fewer distal synapses and the larger $g_{syn}$. Thus, the developmental increase in dendritic field complexity and decreased postsynaptic strength together contribute to the establishment of a two-stage integration model and provide a cellular substrate for a developmental increase in computational power of SCs.

## Discussion

Dendritic integration of synaptic inputs is a critical component of neuronal information processing, but little is known about how it matures during neuronal network formation and maturation. We took advantage of the late development of the cerebellar cortex to characterize developmental changes in synaptic and dendritic properties. By combining patch-clamp recording of cerebellar SC interneurons, 3D reconstructions of their aspiny dendrites, along with the identification of excitatory synapse locations, and numerical simulations, we showed for the first time how the maturation of synapse distribution within interneurons combines with changes in postsynaptic strength and increased dendritic branching to shape the development of neuronal computation. This maturation process favors the emergence of a compartmentalized two-stage integrator model, which extends the repertoire of transformations of synaptic inputs into neuronal output in adult SCs. These results highlight the importance of characterizing not only dendritic morphology, but also synapse placement and synaptic strength, in order to correctly infer a neuron's computational rules.

### Comparison of synaptic properties between immature and adult SCs

Whole-cell voltage-clamp recordings in immature and adult SCs showed a developmental decrease in mEPSC amplitude and slowed kinetics (*Figure 1*). These results contrast with findings in other neurons that show faster mEPSCs during maturation due to changes of AMPAR subunits, vesicular content of neurotransmitter, and/or synapse structure (*Cathala et al., 2005*; *Chen and Regehr, 2000*; *Koike-Tani et al., 2005*; *Yamashita et al., 2003*). Knowing that dendritic inputs could be electrotonically filtered, we took advantage of the ability to selectively stimulate somatic synapses to isolate somatic qEPSCs for comparison between the two ages. Evoked qEPSCs showed a developmental reduction in amplitude (~40%), with no change in kinetics (*Figure 4*). This likely results from the smaller adult PSD size (*Figure 4*), and hence a lower AMPAR number (*Masugi-Tokita et al., 2007*), rather than a developmental reduction in the peak glutamate concentration at the postsynaptic AMPARs (*Figure 1*) or change in AMPAR subunits composition (see Results).

A defining feature of immature SCs is the high propensity of quantal EPSPs to generate spikes (*Carter and Regehr, 2002*). However, the observed developmental decrease in synaptic conductance (*Figure 4*) and increased filtering of mEPSCs (*Figures 1 and 3*) will tend to reduce the influence of single synaptic inputs on somatic voltage in adult SCs, increasing their dynamic range of subthreshold integration. Therefore, individual synaptic inputs are less likely to influence adult SC neuronal output. Moreover, since PSD area is constant along the somatodendritic axis (*Figure 3*), the observed developmental reduction in synaptic conductance can be extrapolated to the whole dendritic tree. Thus, there is no evidence of synaptic conductance scaling mechanisms that offset dendritic filtering, as described for pyramidal neurons (*Katz et al., 2009*; *Magee and Cook, 2000*; *Menon et al., 2013*; *Nicholson et al., 2006*). As a result, SCs show a strong dependence of somatic voltage responses on synapse location within the dendritic arbor.

## Implications of developmental alterations of dendritic morphology

We showed that soon after SC integration into the cerebellar molecular layer microcircuit, their dendrites are nearly the same diameter as adult SCs (*Figure 2*), suggesting a similar capacity for cable filtering of synaptic responses. Since previous studies suggested that SCs were electronically compact (*Carter and Regehr, 2002*; *Llano and Gerschenfeld, 1993*), the observation that mEPSCs from adult SCs were slower led us to consider changes in cable filtering as the underlying mechanism for the developmental change in mEPSC kinetics. However, combining experiments and simulation we showed that immature SCs have similar dendritic integration properties as adult SCs (*Abrahamsson et al., 2012*): (1) dendrite-evoked qEPSCs are smaller and slower than those evoked at the soma (*Figure 3*), consistent with cable filtering (*Figure 2*); (2) STP differed between dendrite and somatic stimulation and did not reveal supralinear recruitment of voltage-gated channels (*Nevian et al., 2007*; *Figure 8*); and (3) simulated subthreshold I/Os were sublinear (*Figure 8*). These results are supported mechanistically by several experimental findings such as (1) the measured membrane time constants that are identical at the two developmental stages, (2) the dendritic diameters being similarly small (*Figure 2*), and (3) the paired-pulse facilitation difference between dendrite and soma which is reproducible with a passive biophysical model like in adult SCs (*Figure 8*). Taken together, the data are consistent with SCs displaying little developmental changes in voltage-gated channel expression (*Molineux et al., 2005*) and low densities of sodium (*Myoga et al., 2009*) and calcium channels (*Tran-Van-Minh et al., 2016*).

The integration of fast synaptic events such as AMPAR-mediated EPSC is critically influenced by the intracellular resistivity, the local membrane capacitance, and dendritic diameter (Equation 2). Since internal resistivity is a challenge to estimate without dual electrode recordings, we examine a wide range of internal resistivities and found that they did not alter qualitatively our conclusions. The narrow dendritic diameters at both ages (*Figure 2B*) suggest that the local input resistance is only slightly different. We cannot rule out that developmental changes in gap junction expression could contribute to the maturation of SC dendritic integration, since they are thought to contribute to the axial resistivity and capacitance of neurons (*Szoboszlay et al., 2016*). All the recordings were made with gap junctions intact, including for membrane time constant measurements. However, their expression in SCs is likely to be lower than their basket cell counterparts (*Hoehne et al., 2020*; *Rieubland et al., 2014*). Overall, the basic electrotonic machinery to filter synaptic responses is already present as soon as SC precursors reach their final location in the outer third of the molecular layer.

However, the developmental differences between mEPSCs, which presumably reflect synaptic responses from the entire dendritic tree, suggested that another factor must be contributing to the difference in apparent electrotonic filtering. Previous studies have shown that synapses are not uniformly distributed along dendrites, allowing pyramidal neuron dendrites to operate as independent computational units (*Katz et al., 2009*; *Menon et al., 2013*; *Polsky et al., 2004*), and retinal starburst amacrine interneurons to compute motion direction (*Vlasits et al., 2016*). We considered the possibility that the distribution of synapse locations within the dendritic tree was altered during SC maturation. We found that synapses were uniformly distributed along the somatodendritic axis with a similar density at the two ages. However, adult SCs had more synapses located at further electrotonic distances (~2/3 vs. 1/3 of synapses were more than 30 µm from the soma; *Figures 6 and 7*) due to increased dendritic branching. Thus, the distal-weighted synaptic distribution in adult SCs favors inputs that experience stronger cable filtering. This was confirmed by simulating a mean mEPSC at the two ages, that fully reproduce the mEPSC recorded experimentally (*Figure 1*), by weighting simulated qEPSCs according to the relative number of synapses at specific distances along the dendrites (*Figure 7*). Indeed, the large fraction of distal synapses in adult SCs was sufficient to account for the observed developmental difference in mEPSC amplitude and time-course. Moreover, since the space constant does not change significantly with development and the dendritic tree complexity increases, the number of computational segments is expected to increase with age.

## Developmental changes in computational rules

Our findings highlight the critical importance of understanding both the structural and functional mechanisms underlying developmental refinement of synaptic integration that drives a neuron's computational properties and, the emergence of mature microcircuit function. While a defining feature of immature SCs is the high propensity of quantal EPSPs to generate spikes (*Carter and Regehr, 2002*),

the observed developmental decrease in synaptic conductance (*Figure 4*) and increased filtering of mEPSCs (*Figures 2 and 3*) will tend to reduce the influence of single synaptic inputs on somatic voltage in adult SCs, increasing their dynamic range of subthreshold integration. Although dendrites in immature and adult SCs exhibit similar electrotonic filtering, the distal bias in synapse location promotes sublinear subthreshold dendritic integration in adult SCs. Unlike pyramidal neurons, where synapse strength and density are scaled to normalize the contribution of individual inputs to neuronal output (*Katz et al., 2009*; *Magee and Cook, 2000*; *Menon et al., 2013*), the spatially uniform distribution of synapse strength and density in SCs do not compensate the electrotonic filtering effects of the dendrites or the increased number of distal synapses due to branching.

These properties of quantal synaptic responses, together with the larger difference in sublinearity between soma and dendrites (*Figure 8D*), will favor the emergence of a spatially compartmentalized two-stage integration model in adult SCs, thereby promoting location-dependent integration within dendritic subunits (*Polsky et al., 2004*) and enhanced neuronal computations (*Cazé et al., 2013*). In immature SCs, the repertoire of computations is more similar to a simple single-stage integration model where large and fast synaptic potentials will promote reliable and precise EPSP–spike coupling (*Cathala et al., 2003*; *Fricker and Miles, 2001*; *Hu et al., 2010*), which may be critical for driving the functional maturation of the local microcircuit (*Akgül and McBain, 2020*). In contrast, synaptic integration and summation in adult SCs can obey different rules depending on synapse location within the dendritic tree enabling to discriminate a larger number of spatial patterns of synaptic activation (*Tran-Van-Minh et al., 2015*) and therefore favor spatially sparse synaptic representations (*Abrahamsson et al., 2012*; *Cazé et al., 2013*) that might be essential for the development of enhanced pattern separation by Purkinje cells (*Cayco-Gajic et al., 2017*). Since a recent theoretical study showed that sublinear integration is also a property of hippocampal fast-spiking interneurons (*Tzilivaki et al., 2019*) that influence memory storage, it will also be important to determine if these interneurons exhibit a similar maturation of their neuronal computation.

## Implications for neurodevelopmental and neurological disorders

The increasing complexity of dendritic arbors, accompanying changes in synaptic connectivity and function during development is not limited to the cerebellum. These maturational processes are altered in neurodevelopmental disorders, such as mental retardation (*Kaufmann and Moser, 2000*), autism spectrum disorders (*Antoine et al., 2019*; *Peng et al., 2016*), or Rett syndrome (*Blackman et al., 2012*; *Ip et al., 2018*), as well as in neurodegenerative disease. Indeed, these developmental processes are particularly relevant for interneurons since they play a pivotal role in the establishment of the correct excitation/inhibition balance for normal circuit function. During development, inhibitory interneurons are essential for defining critical periods (*Gu et al., 2016*; *Hensch et al., 1998*) or direction selectivity in the retina (*Vlasits et al., 2016*; *Wei et al., 2011*), so that interneuron dysfunction is associated with neurodevelopment disorders (*Akerman and Cline, 2007*; *Le Magueresse and Monyer, 2013*; *Marín, 2016*). Our work demonstrates how developmental changes in neuronal morphology, and synapse distribution and strength, combine to determine the impact of synaptic inputs on neuronal output. Our findings provide a functional template of how dendritic integration matures throughout development to enrich interneurons with more complex neuronal computations, promoting location-dependent integration within dendritic subunits.

## Materials and methods

### Key resources table

| Reagent type (species) or resource | Designation | Source or reference | Identifiers | Additional information |
|---|---|---|---|---|
| Strain, strain background (*Mus musculus* – male and female) | CB6F1 | Mouse Genome Informatics | RRID:MGI:5649749 | |

*Continued on next page*

*Continued*

| Reagent type (species) or resource | Designation | Source or reference | Identifiers | Additional information |
|---|---|---|---|---|
| Strain, strain background (*Mus musculus* – male and female) | PSD-95-CreNABLED mice | Haining Zhong, Vollum Institute | DOI: https://doi.org/10.1523/JNEUROSCI. 3888-14.2014 | |
| Strain, strain background (*Mus musculus* – male and female) | B6.129-*Nos1*[tm1(cre)Mgmj]/J | Mouse Genome Informatics | https://www.jax.org/strain/017526 | |
| Chemical compound, drug | Gabazine (SR-95531) | Abcam | Cat#:ab120042 | |
| Chemical compound, drug | QX-314 | Abcam | Cat#:ab120118 | |
| Chemical compound, drug | 7-Chlorokynurenic acid | Abcam | Cat#:ab120255 | |
| Chemical compound, drug | Alexa Fluor 488 | ThermoFisher Scientific | Cat#:A10436 | |
| Chemical compound, drug | Alexa Fluor 594 | ThermoFisher Scientific | Cat#:A10438 | |
| Chemical compound, drug | D-AP5 | Abcam | Cat#:ab120003 | |
| Chemical compound, drug | Gamma DGG | Abcam | Cat#:ab120307 | |
| Chemical compound, drug | TTX | Abcam | Cat#:Ab120055 | |
| Chemical compound, drug | Strychnine | Sigma | Cat#:S0532 | |
| Software, algorithm | Reconstruct | JC Fiala | DOI: 10.1111 /j.1365–2818.2005.01466 .x | |
| Software, algorithm | Igor Pro | Wavemetrics | RRID:SCR_000325 | |
| Software, algorithm | Neuromatic | *Rothman and Silver, 2018*; DOI:10.3389 | RRID:SCR_004186 | |
| Software, algorithm | GraphPad Prism 6 | GraphPad Software | RRID:SCR_002798 | |
| Software, algorithm | ImageJ | National Institutes of Health | RRID:SCR_003070 | |
| Software, algorithm | Imaris | Bitplane Imaris Software | RRID:SCR_007370 | |
| Software, algorithm | Zeiss Atlas | Zeiss | https://www.zeiss.com/microscopy/int/ products/microscope-software/atlas.html | |

## Slice preparation and electrophysiology

Acute cerebellar parasagittal slices (250 or 200 µm thick, respectively) were prepared from immature (postnatal day P13–19) and adult (P35–57) mice (F1 of BalbC and C57B6 or C57BL/6 J) as described previously (*Abrahamsson et al., 2012*). Animal housing and all procedures were approved by the Comité National d'Ethique pour les Sciences de la Vie et de la Santé (Comité Charles Darwin, authorisation no. 1492-02) and the Ethics Committee #89 of Institut Pasteur (CETEA; protocol approval #DHA180006). Briefly, mice were killed by decapitation, the brains rapidly removed and placed in an ice-cold solution containing (in mM): 2.5 KCl, 0.5 $CaCl_2$, 4 $MgCl_2$, 1.25 $NaH_2PO_4$, 24 $NaHCO_3$, 25 glucose, and 230 sucrose, bubbled with 95 % $O_2$ and 5 % $CO_2$. Slices were cut from the dissected cerebellar vermis using a vibratome (Leica VT 1000 S or VT1200S), incubated at 32 °C for 30 min in the following solution (in mM): 85 NaCl, 2.5 KCl, 0.5 $CaCl_2$, 4 $MgCl_2$, 1.25 $NaH_2PO_4$, 24 $NaHCO_3$, 25 glucose, and 75 sucrose and subsequently maintained at room temperature for up to 8 hr in the recording solution containing (in mM): 125 NaCl, 2.5 KCl, 2 $CaCl_2$, 1 $MgCl_2$, 1.25 $NaH_2PO_4$, 25

NaHCO₃, and 25 glucose. Unless otherwise noted, this solution included during patch recordings 10 µM SR-95531, 10 µM D-AP5, 20 µM 7-chlorokynurenic acid, and 0.3 µM strychnine, to block GABA_A, NMDA, and glycine receptors, respectively.

Whole-cell patch-clamp recordings were made from SCs located in the outer one-third at molecular layer at temperatures ranging from 33 to 36°C using an Axopatch-200A or a Multiclamp 700 amplifier (Axon Instruments, Foster City, CA, USA) with fire-polished thick-walled glass patch-electrodes (tip resistances of 6–8 MΩ) that were backfilled with a solution containing (in mM): 117 K-MeSO₄, 40 HEPES, 6 NaOH, 5 EGTA, 1.78 CaCl₂, 4 MgCl₂, 1 QX-314-Cl, 0.3 NaGTP, 4 NaATP, and, when applied 0.03 Alexa 594, adjusted to ~300 mΩ and pH 7.3. Series resistance and capacitance measures were determined directly from the amplifier settings.

All EPSCs were recorded at −70 mV holding membrane potential (not corrected for LJP ~ +6 mV), were filtered at 10 kHz, and digitized at 100 kHz using an analog-to-digital converter (model NI USB 6259, National Instruments, Austin, TX, USA) and acquired with Nclamp (*Rothman and Silver, 2018*) within the Igor Pro 6.2 environment, WaveMetrics. Series resistance was on average of 16 mΩ, uncompensated, and monitored throughout the experiment. Recordings were discarded if series resistance exceeded 20 mΩ. To evoke EPSC, PFs were stimulated with a glass patch electrode filled with external recording solution that was placed close to a fluorescently labeled dendrite or close to the soma. 50 µs pulses between 5 and 55 V (Digitimer Ltd, Letchworth Garden City, UK) were delivered as described previously (*Abrahamsson et al., 2012*). Somatic and dendritic quantal EPSCs were obtained from experiments where [Ca²⁺] was lowered to 0.5 mM while [Mg²⁺] was increased to 5 mM to obtain an evoked EPSC success rate <10% known to produce a qEPSC with a <10% amplitude error (*Silver, 2003*). Trials with a synaptic event could be clearly selected by eye. The stimulation artifact was removed by subtracting from single success traces the average obtained for the traces with failed synaptic transmission.

Current-clamp recordings were performed using a Multiclamp 700 amplifier. Patch electrodes were coated with dental wax and series resistance was compensated by balancing the bridge and compensating pipette capacitance. Current was injected to maintain a holding membrane potential near −70 mV. Data were filtered at 10 kHz, and digitized at 100 kHz.

D-AP5, 7-chlorokynurenic acid, γDGG, QX-314 chloride, SR 95531, and tetrodotoxin were purchased from Ascent Scientific (http://www.ascentscientific.com). Alexa 594 and Alexa 488 were purchased from Invitrogen (https://www.thermofisher.com/invitrogen). All other drugs and chemicals were purchased from Sigma-Aldrich (https://www.sigmaaldrich.com).

## Multicompartmental biophysical modeling

Passive cable simulations of EPSC and EPSP propagation within idealized and reconstructed SC models were performed using Neuron 7.1, 7.2, and 7.5 (*Hines and Carnevale, 1997*). The idealized SC model had a soma diameter of 9 µm and three 90 µm long dendrites of 0.47 µm diameter, with either one, three, or five branches. An immature (P16) and adult SC (P42) were patch loaded with 30 µM Alexa 594 in the pipette and imaged using 2PLSM. Both cells were reconstructed in 3D using NeuronStudio in a semiautomatic mode which uses a robust subpixel estimation algorithm (calculation of Rayburst diameter [*Rodriguez et al., 2008*]). We manually curated the diameters to verify that it matched the fluorescence image to faithfully account for all variations in diameter throughout the dendritic tree. The measured diameter across the entire dendritic tree of the reconstructed immature and adult SCs was 0.42 and 0.36 µm, respectively. The 16 % smaller diameter in adult was similar to the 13 % obtained from confocal image analysis from many SCs (see *Figure 2B*).

The 3D reconstruction was then imported into NEURON. Passive properties were assumed uniform across the cell. Specific membrane capacitance ($C_m$) was set to 0.9 µF/cm². $R_m$ was set at 20,000 Ω.cm² to produce in the SC models a membrane time constant matching the one measured experimentally: 19 ± 2.2 ms for immature SCs (*n* = 16) and 17 ± 2.7 ms for adult SC (*n* = 10). $R_i$ was set to 150.Ω cm to match the filtering of EPSC decay in the dendrites of mature SC (*Abrahamsson et al., 2012*) and allowed to vary from 100 to 200 Ω cm to sample a large range of physiological $R_i$ since its physiological value is not known. The peak and kinetics of the AMPAR-mediated synaptic conductance waveforms ($g_{syn}$) were set to simulate qEPSCs that matched the amplitude and kinetics of experimental somatic quantal EPSCs and evoked EPSCs. Immature quantal $g_{syn}$ had an peak amplitude of 0.00175 µS, a 10–90% RT of 0.0748 ms and a half-width of 0.36 ms (NEURON synaptic conductance parameter Tau0

= 0.073 ms, Tau1 = 0.26 ms, and Gmax = 0.004 µS) while mature quantal $g_{syn}$ had an peak amplitude of 0.00133 µS, a 10–90% RT of 0.072 ms and a half-width of 0.341 ms (NEURON synaptic conductance parameters Tau0 = 0.072 ms, Tau1 = 0.24 ms, and Gmax = 0.0032 µS). For all simulations, the reversal potential was set to 0 mV and the holing membrane potential was set at −70 mV. Experimental somatic PPR for EPSCs was reproduced with a $g_{syn}2/g_{syn}1$ of 2.25.

## Electron microscopy and 3D reconstructions

Electron microscopy and 3D reconstructions of two SCs from acute slices (postnatal days 14 and 17) were performed as described previously (*Abrahamsson et al., 2012*). Slices containing SCs whole-cell patched with a K-MeSO3-based internal solution containing biocytin (0.3%) and Alexa 594 (30 µM) were transferred to a fixative containing paraformaldehyde (2.5%), glutaraldehyde (1.25%), and picric acid (0.2%) in phosphate buffer (PB, 0.1 M, pH = 7.3), and fixed overnight at room temperature. After washing in PB, slices were transferred to sucrose solutions (15 % for 30 min, then stored in 30%) for cryoprotection and frozen in liquid nitrogen, then subsequently thawed. The freeze–thaw cycle was repeated twice, then followed by incubation with a 1.4 nm gold-conjugated streptavidin (Nanoprobe, 1:100 in Tris-buffered saline [TBS] and 0.03 % Triton X100). After washing in TBS and dH$_2$O, slices were treated with HQ silver enhancement kit (Nanoprobe) for 5 min, fixed in 1 % OsO4 in PB for 30 min, and block stained with 1 % uranyl acetate for 40 min. After dehydration through a series of ethanol solutions (50%, 70%, 80%, 90%, 95%, 99%, and 100%) and propylene oxide twice for 10 min, slices were embedded into Durcupan (Fluka) and flat embedded. The labeled SCs were trimmed and 300–400 serial ultrathin sections were cut at 70 nm using ATUMtome (RMC Boeckeler). Serial sections containing immunogold labeled profiles were imaged with a scanning electron microscope (Merlin Compact, Zeiss) and Zeiss Atlas package at ×22,000 for whole-cell reconstruction and at ×55,000 for synapses. In order to compare synapse area measurement comparable to those done on adult soma (*Abrahamsson et al., 2012*), we performed serial sections and then standard transmission electron microscopy (TEM) on three unlabeled SCs (which avoids potential distortions due to the patching). These sections were cut at 70 nm using an ultramicrotome (Leica EM UC7), observed with a transmission electron microscope (Tecnai 12, FEI), and photographed at ×21,000. Asymmetrical synapses made by axon terminals onto SC somata and dendrites were analyzed only if they were fully present within the serial sections. The PSD length of the asymmetrical synaptic membrane specialization was measured on each ultrathin section, and the PSD area was calculated by multiplying the summed synaptic length from each synapse with the thickness (70 nm) of the ultrathin sections. The 3D reconstruction of the two SC soma and parts of their dendritic trees was performed using the software Reconstruct (JC Fiala). The distances from each synapse to the soma were measured along the dendrites in the reconstructed volume. In order to compare PSD area to those in adult SC, we only considered those measurements using the same TEM method (*Abrahamsson et al., 2012*).

## Transmitted light and fluorescence imaging

SC somata in the outer one-third of the molecular layer were identified and whole-cell patched using infrared Dodt contrast (Luigs and Neumann). LED illumination or in some cases two-photon excitation, coupled with the Dodt contrast, were used to visualize Alexa 594-filled SCs and position extracellular stimulating electrodes along isolated dendrites of SCs fluorescence. Two-photon excitation was performed with a pulsed Ti:Sapphire laser (MaiTai DeepSee, Spectra Physics) tuned to 810 nm and images were acquired with an Ultima two-photon laser scanning microscope system (Bruker) mounted on an Olympus BX61WI microscope equipped with a ×60 (1.1 NA) water-immersion objective. LED excitation (470 nm) was performed with a CAIRN LED module (optoLED) and wide-field fluorescence images were acquired with a CCD camera (QIclick, QImaging) mounted on an Olympus BX51 microscope equipped with a ×60 (1 NA) water-immersion objective.

One-photon confocal laser scanning fluorescence microscopy was performed with an Ultima scan head mounted on a Nikon EFN microscope. SCs were filled with 40 µM Alexa 488. Maximal intensity projections of confocal images were performed using a ×100 1.1 NA Nikon dipping objective in 0.2 µm increments as described previously (*Abrahamsson et al., 2012*). Dendritic diameters were estimated from the FWHM of pixel intensity line profiles on 1 µm segments of dendrites, made perpendicular to dendritic length using the same procedure as for adult SCs (*Abrahamsson et al., 2012*). In brief, we selected dendritic segments for measurement throughout the dendritic tree to

homogeneously sample the diversity of diameters. We collected 430 line profiles from 18 immature SCs, then averaged the profiles from the same dendritic segments ($n$ = 93). The imaging resolution within the molecular layer was estimated from the width of intensity line profiles of SC axons. The FWHM was 0.30 ± 0.01 μm ($n$ = 57 measurements over 16 axons) and a mean of 0.27 ± 0.01 μm ($n$ = 16) when taking into account the thinnest section for each axon. Only 2 % of all dendritic measurements are <270 nm, suggesting that the dendritic diameter estimation is hardly affected by the resolution of our microscope.

2 P imaging: The two-photon scanning microscope (2PLSM, Ultima IV, Bruker) was equipped with a Ti:Sapphire Laser (Chameleon II, Coherent Inc) at 940 nm (SC body, Alexa 594) and 810 nm (Venus-tagged PSD95 puncta) using a ×60 water-immersion objective (LUMFL N, 1.10 NA, Olympus). The point spread function of the microscope was estimated from the FWHM value of the x-, y-, and z-intensity line profiles from 3D images of 200 nm yellow-green fluorescent latex beads (FluoSpheres, F8811, Thermo Fisher Scientific): $PSF_{810x/y}$ = 325 ± 27 (SD) nm, $PSF_{810z}$ = 1178 ± 121 (SD) nm, and $PSF_{940x/y}$ = 390 ± 23 (SD), and $PSF_{940z}$ = 1412 ± 141 (SD) nm.

## 3D reconstructions and puncta detection from 2P images

We examined the distribution of excitatory synaptic inputs along the somatodendritic compartment in SCs from a transgenic mouse line that conditionally expresses Venus-tagged PSD95 under the control of the nitric oxide synthase one promoter (PSD95-Enabled [*Fortin et al., 2014*] × Nos1 Cre [*Kim et al., 2014*]). We patch-loaded single SCs with the fluorescence indicator Alexa 594, then performed live two-color 2PLSM to identify Venus-tagged PSD95 puncta associated with the labeled somata and dendritic trees. *Z*-Stacks were acquired for each wavelength with a *z*-step of 300 nm, a pixel size of 154 nm, and an image size of 512 × 512 pixels. To correct for a shift in the focal point for the different wavelengths and a potential drift in *x/y*-directions, the individual stacks were registered to each other using as a reference the dendritic structure imaged using Alexa 594 emission, which is primarily excited at 810 nm, but weakly excited at 940 nm which allows to record both the puncta and the cell body simultaneously. The registration was performed using the IMARIS *stitcher* tool. Fluorescence emission was spectrally separated from laser excitation using a custom multipass dichroic (zt 405/473–488/nir-trans; Chroma) and a short pass IR blocking filter (ET680sp-2p8; Chroma). Venus and Alexa 594 fluorescence emission were spectrally separated and detected using detection filter cubes consisting of a long-pass dichroic (575dcxr; Chroma) and two bandpass filters (HQ525/70 m-2p and HQ607/45 m-2p, respectively; Chroma). A multialkali (R3896, Hamamatsu, Japan) photomultiplier tubes was used to detect Alexa 594 fluorescence and gallium arsenide phosphide tube (H7422PA-40 SEL, Hamamatsu) for the Venus channel. Proximal and substage detection were used to increase signal to noise.

Dendrite tracing: The image analysis software IMARIS 9.5 (Bitplane) was used for dendritic tracing and fluorescence puncta detection. Fluorescence images were filtered using a 3 × 3 px median filter to remove noise. Image stacks were then further combined using the IMARIS *stitcher* tool to create one contiguous file that permits tracing of the entire dendritic tree. Dendrites, but not axons, were traced in IMARIS using the *filament* tool in a semiautomatic mode using the *AutoPath* method and the options *AutoCenter* and *AutoDiameter* activated. The soma center was chosen as the starting point with centripetally tracing along dendrites. Dendritic lengths were estimated as the dendritic path distance from the center of the soma and Sholl analysis was performed with IMARIS from all reconstructed SC.

PSD-95 Venus puncta detection: Fluorescence puncta were detected using the IMARIS spot creation tool, using background fluorescence subtraction to compensate for different levels of fluorescent intensity along the *z*-axis of the stack. The initial minimal spot search size was set to 300 × 300 × 1100 nm, slightly smaller than the PSF at 810 nm. No further thresholds or criteria were applied inside IMARIS. Parameters describing the fluorescent puncta, including the intensity at the center, their spatial coordinates, and their diameters, were exported as excel files via the Statistics tab and further analyzed using custom python scripts. In order to separate the PSD95 puncta from false detection of noise, two threshold criteria were applied. (1) All spots with a diameter smaller than and equal to the PSF (300 × 300 × 1100 nm) were rejected. (2) Only spots with a peak intensity larger than the mean of the background intensity plus three times its standard deviation of the background noise, were considered for subsequent analysis. In combination, the thresholds ensure that the spots

originated from fluorescent puncta and not false positives generated from noise fluctuations. The background intensity and its corresponding standard deviation were measured for each file in *Fiji*, by selecting regions without puncta and using the *Measure* tool to calculate the mean and standard deviation. In order to ensure consistent sampling of the background, mean and noise estimates were made from at least ten different regions at different *z* positions in each stack, corresponding to an area between 17–27 $\mu m^2$ (790–1210 pixels) and 10–40 $\mu m^2$ (430–1850 pixels), for the immature and adult SCs, respectively. This approach is limited by the resolution of 2 P fluorescence imaging to differentiate individual synapses within clusters and thus may result in an underestimate of absolute synapse density, but allowed for an unbiased estimate of synapse distributions at the two developmental stages.

Puncta located on somata were selected from the total pool of detected puncta (described above) using the following criteria: (1) spots were associated with the soma if the peak intensity at the position of the spot center in the Alexa 594 channel was larger than half the maximum of the whole stack, (2) detected spot diameters were greater than the size PSF (300 × 300 × 1100 nm), and (3) spot intensities were larger than the mean plus 3*SD of background intensity of the PSD95-Venus channel, measured from within the soma. Puncta from somata that showed saturation in the Alexa 594 channel were not included in the analysis.

Analysis of spot and dendrite distances: The structure of the dendritic tree, as well as the position of the puncta, were further analyzed using custom python scripts (Rückerl F.2021. DevNonlinSynIntCIN. Github. https://github.com/Unit-of-Synapse-and-Circuit-Dynamics/DevNonlinSynIntCIN (*Rückerl, 2021*) copy archived at swh:1:rev:04b0c50f05a67a008e52c515b0fb21bad1e2f5e5). The dendritic tree was reconstructed with the center of the soma as its root using the python *NetworkX* package. Fluorescence puncta were considered to arise from the labeled dendrite if they were located within a maximal distance from the center of the dendrite. This distance was taken as the dendritic radius, estimated from IMARIS, plus 200 nm (~HWHM of the $PSF_{810}$). The estimation of local radius was made from IMARIS binary masks using a threshold of the local contrast (*DiameterContrastThreshold*) set at three times the standard deviation above the background fluorescence noise. The diameter is then calculated using the *Shortest Distance from Distance Map* algorithm, which estimates the diameter as the distance from the center of the dendrite to the closest part of the surface determined by the above threshold. The average dendritic radius, using this approach, was found to be 0.66 ± 0.28 (SD) μm for adult, and 0.72 ± 0.27 (SD) μm for immature mice. As this value was larger than that estimated from single-photon confocal imaging, it was only used as a part of the criteria for assigning a fluorescence puncta to a reconstructed dendrite. For each dendritic branch, the number of PSD95 puncta and their distance to the soma surface were calculated. As the data points of the dendrite structure obtained from IMARIS are not homogeneously spaced, the dendritic structure was resampled in 100 nm intervals with the distance for each segment recalculated with respect to the starting point of each corresponding branch from the soma surface (estimated using the binary mask as for the dendrites). Histograms for the distribution of PSD95 puncta and the number of dendritic segments at a given distance from the soma were then generated in 10 μm bins and used to estimate the puncta density along the dendritic tree. Cumulative plots were sampled at 1 μm intervals.

## Electrophysiology analysis

Data analysis was performed using the Neuromatic analysis package (*Rothman and Silver, 2018*) written within the Igor Pro environment (WaveMetrics, Lake Oswego, OR, USA). mEPSCs were detected with a threshold detection method and mEPSC population average is calculated from the mean EPSC response calculated for each SC. All EPSCs were baseline subtracted using a 1 ms window before the stimulation artifact. Peak amplitudes were measured as the difference between the baseline level immediately preceding the stimulation artifact, and the mean amplitude over a 100 μs window centered on the peak of the response. EPSC rise time was estimated as the time the EPSC rose from 10 to 90% of the peak amplitude (in ms). Decay kinetics were assessed either as the width of the EPSC at the amplitude one-half of the peak (half-width in ms) or as the weighted time constant of decay ($\tau_{decay}$) calculated from the integral of the current from the peak, according to:

$$\tau_{decay} = \frac{\int_{t_{peak}}^{t_{\infty}} I(t)dt}{I_{\text{peak}}}$$

where $t_{peak}$ is the time of the EPSC peak, $t_{\infty}$ is the time at which the current had returned to the pre-event baseline, and $I_{\text{peak}}$ the peak amplitude of the EPSC.

All data are expressed as average ± SEM otherwise noted. Statistical tests were performed using a nonparametric Wilcoxon–Mann–Whitney two-sample rank test routine for unpaired or a Wilcoxon signed-rank test routine for paired comparisons, unless otherwise stated. Linear correlations were determined using a spearman test. Unless otherwise noted, unpaired tests were used and considered significant at p < 0.05 (OriginPro, Northampton, MA, USA).

Equation 1. Length constant for an infinite cable.

$$\lambda_{DC} = \sqrt{\frac{dR_{\text{m}}}{4R_{\text{i}}}},$$

where $d$ is the dendritic diameter and $R_{\text{m}}$ and $R_{\text{i}}$ are the specific resistance of the membrane and internal resistivity, respectively.

Equation 2. Frequency-dependent length constant for an infinite cable.

$$\lambda_{\text{AC}} = \lambda_{\text{DC}} \sqrt{\frac{2}{1+\sqrt{1+(2\pi f\tau_m)^2}}},$$

where $f$ is the frequency representing an AMPAR current and $\tau_{\text{m}}$ is the membrane time constant. When $f$ is greater than 100 Hz, this can be simplied when $f$ is greater than 100 Hz:

$$\lambda_{AC} \approx \sqrt{\frac{d}{4\pi f R_{\text{i}} C_{\text{m}}}}$$

# Acknowledgements

This study was supported by the Centre National de la Recherche Scientifique and the Agence Nationale de la Recherche (ANR-13-BSV4-00166, to LC and DAD). TA was supported by fellowships from the Fondation pour la Recherche Medicale and the Swedish Research Council. We thank Dmitry Ershov from the Image Analysis Hub of the Institut Pasteur, Elodie Le Monnier, Elena Hollergschwandtner, Vanessa Zheden, and Corinne Nantet for technical support and Haining Zhong for providing the Venus-tagged PSD95 mouse line. We would like to thank Alberto Bacci, Ann Lohof, and Nelson Rebola for comments on the manuscript.

# Additional information

## Funding

| Funder | Grant reference number | Author |
| --- | --- | --- |
| Agence Nationale de la Recherche | ANR-13-BSV4-00166 | Laurence Cathala David DiGregorio |
| Agence Nationale de la Recherche | ANR-18-CE16-0018-03 | David A DiGregorio |
| Agence Nationale de la Recherche | ANR-19-CE16-0019-02 | David A DiGregorio |

The funders had no role in study design, data collection and interpretation, or the decision to submit the work for publication.

## Author contributions

Celia Biane, Formal analysis, Investigation, CB performed electrophysiology experiments and analysis; Florian Rückerl, Data curation, Formal analysis, Visualization, Writing – review and editing, FR and DAD designed PSD95-venus experiments. FR performed the analysis of PSD95 puncta; Therese

Abrahamsson, Formal analysis, Investigation, Writing – review and editing; Cécile Saint-Cloment, Resources, CS-C was in change of the Nos1-PSD95 mouse management; Jean Mariani, Rachel M Sherrard, Funding acquisition, Writing – review and editing; Ryuichi Shigemoto, Data curation, Formal analysis, Validation, Investigation, Methodology, Writing – review and editing; David A DiGregorio, Data curation, Supervision, Funding acquisition, Validation, Project administration, Writing – review and editing; Laurence Cathala, Conceptualization, Data curation, Software, Formal analysis, Software, Funding acquisition, Validation, Investigation, Methodology, Writing – original draft, Project adminis-tration, Writing – review and editing, Visualization, LC designed all experiments except PSD95-venus experiments, which were designed by FR and DAD. LC performed electrophysiology experiments and analysis, confocal and 2P imaging, confocal images analysis, reconstruction in Neuronstudio and numerical simulations. LC wrote the manuscript and all authors edited it

**Author ORCIDs**
Ryuichi Shigemoto http://orcid.org/0000-0001-8761-9444
David A DiGregorio http://orcid.org/0000-0002-6417-4566
Laurence Cathala http://orcid.org/0000-0001-6253-1390

**Ethics**
Animal housing and all procedures were approved by the Comité National d'Ethique pour les Sciences de la Vie et de la Santé (Comité Charles Darwin, authorisation N° 1492-02) and the Ethics Committee #89 of Institut Pasteur (CETEA; protocol approval #DHA180006).

**Decision letter and Author response**
Decision letter https://doi.org/10.7554/eLife.65954.sa1
Author response https://doi.org/10.7554/eLife.65954.sa2

---

## Additional files

**Supplementary files**
• Transparent reporting form

**Data availability**
All data generated or analysed during this study are included in the manuscript and supporting files. Source data files have been provided for Figures 1,2,3,4,6,7 and 8.

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
