## [Decision Letter]

**Acceptance summary:**

Several properties that affect neuronal function change over the course of cell maturation, including morphology, membrane properties and synaptic connectivity. Here the authors delineate the contribution of these parameters to synaptic integration in developing cerebellar stellate cells. Using a range of cutting edge approaches, they report that changes in synaptic distribution dominate the observed maturation of dendritic integration, providing new insight into the mechanisms contributing to neuronal maturation.

**Decision letter after peer review:**

Thank you for submitting your article "Developmental emergence of two-stage nonlinear synaptic integration in cerebellar interneurons" for consideration by *eLife*. Your article has been reviewed by 3 peer reviewers, one of whom is a member of our Board of Reviewing Editors, and the evaluation has been overseen by John Huguenard as the Senior Editor. The reviewers have opted to remain anonymous.

Essential revisions:

1. The authors should provide additional verification that the biophysical parameters of the model built for adult SCs in Abrahamsson et al., (2012) are appropriate for cells from younger animals and address the potential contribution of developmental changes in AMPAR/synaptic properties.

2. Broaden the scope of the conclusions by exploring additional implications of the observed developmental changes, for example, by testing whether inclusion of NMDARs alters the relative strength of distal inputs, how gap junction coupling between SCs would affect the results, and testing different dendritic impedance scenarios.

3). Provide all the technical details that would allow reproduction of the experiments. More methodological information for both the electrophysiology and simulations need to be included, including series resistance errors.

*Reviewer #1 (Recommendations for the authors):*

1) Line 150-158. Foster et al., 2005 shouldn't be used to support exclusive uni-vesicular release at PF-MLI synapses, since it addresses multi-vesicular release and AMPAR saturation at PF-PC synapses. In general, a central assumption that mEPSCs provide an unbiased assessment of the effective synaptic strength is not convincing because there is abundant evidence that PF-MLI synaptic contacts are heterogeneous in terms of the number of vesicles released per action potential (Malagon et al., 2016; Pulido and Marty, 2019; Vaden et al., 2019). Nahir and Jahr (2013) provide support that a majority of evoked release is mediated by a single vesicle at PF-MLI synapses in adolescent rats, but considering the goals of this manuscript to assess the contribution of 'synaptic strength" over development, the authors should address whether this variable contributes to the differences between mEPSC and qEPSCs as well as qEPSCs over development.

2) Figure 2. It looks like the amplitude of the simulated EPSC drops off faster as a function of distance in the mature vs. immature SC (these data should be presented on the same scale). Is this due to increased branching in the mature cell that in turn mediates the difference in Figure 8D? Please discuss more explicitly.

3) Figure 7. It would strengthen the conclusion to see this model validated experimentally with electrical stimulation, such as by plotting the qEPSC amplitudes and kinetics as a function of distance of the stim electrode from the soma. It seems like these data might already exist, but aren't presented directly. Including experimental data like what is shown in Figure 1 and 2 of Abrahammson et al., but in the adolescent SCs would validate that the previously established model still applies in cells from younger animals, and help rule out other contributions from AMPARs or dendritic membrane properties, or synaptic glutamate concentration (see pt 1).

4) Figure 8 B,C,D. It is not surprising that the same biophysical model with smaller dendritic branches recapitulates the behavior of the adult SC with minor quantitative differences (it is hard to imagine a different outcome). But one wonders whether this simulation can be verified experimentally or perhaps more importantly, whether conclusions would be generalizable if NMDAR activation (which seems likely to change over maturation) is considered. The exclusive focus on AMPAR-only integration seems of limited relevance for understanding the computational changes occurring over SC development.

*Reviewer #2 (Recommendations for the authors):*

P. 7 l. 197 the different sections of the dendrites per cell should have been quantified separately to determine whether there are variations among dendrites in the same cell and among different cells. The question of how locations for diameter measurements (see also the next point) were chosen is relevant in particular when adult SCs presenting higher dendritic complexity reflected in more branches are being compared to immature SCs. It would be worth considering to break down the diameters per branch order, instead of collating all data in a global mean (see below P15, l. 429-430). All this could affect the properties of dendrites as well as the distribution of synapses along the dendrites (see Figure 2B p33).

P. 31 Figure 2A,B how did the authors ensure fair sampling of the dendritic tree for the measurement of dendritic diameters?

P. 31 Figure 2B Compare to Abrahamsson et al., 2012 Figure 4B: it would be helpful to see the two distributions (immature and mature SC) together.

P. 9 l. 280 "PSD area was constant…." But in Figure 3G there is no quantification of this. It is just an observation. Provide quantitative data about PSD are at the soma vs dendrites and distance from the soma, as well as cumulative plots and bin distributions instead of Figure 3G (as done for LM data in Figure 6).

P. 10 l. 298-299 n = 83 from 3 cells (see Figure 4C legend) vs n = 97 from 2 cells from Abrahamsson et al., 2012. N = 83 from 3 cells from immature when n=97 from 2 cells in mature: is the number of excitatory synapses onto the soma the same in immature vs mature SC? A density measurement would be more appropriate to compare variations, if any.

P. 9 l. 277: Why did the authors not compare (and pool if statistically possible) the data from the 3 unloaded SCs (Figure 4C) and the 2 loaded cells (used in Figure 3F)?

P. 10 l. 306-311 "In addition to cable filtering of synaptic currents, a neuron's somatic response to dendritic input is also affected by synapse distribution within its dendritic tree,….." and P. 10 l. 320-321 "To test this hypothesis, we examined the distribution of excitatory synaptic inputs along the somato-dendritic compartment in immature and adult SCs." Light microscopy was utilized to quantify these parameters (Figure 6). These parameters should also be quantifiable from the EM data the authors already have. From these data it should be possible to quantify and plot "synapse number vs distance from the soma". These data should then be compared to the data presented at P. 11 l. 326-336, as well as to the previous EM data about PSD size once these have also been quantified (P. 9 l. 280). EM data should accompany the LM data even more so when the authors themselves point out in Methods (l. 719-722) "This approach is limited by the resolution of 2P fluorescence imaging to differentiate individual synapses within clusters and thus may result in an underestimate of absolute synapse density, but allowed for an unbiased estimate of synapse distributions at the two developmental stages."

P. 11 l. 323-324 "We then mapped Venus-tagged PSD95 puncta associated with the somata and dendritic trees of Alexa 594-filled immature and adult SCs (Figure 5A-D)" and l. 326-328 "Venus-tagged PSD95 puncta within the soma and dendritic tree of 9 immature and 8 adult 3D-reconstructed SCs (Figure 5G), showed ~ 80 % more puncta on adult SCs (582 {plus minus} 48 puncta vs. 324 {plus minus} 21 in immature SC, p<0.05)."

No data about the PSD95 puncta on somata only are presented (i.e. separated from the puncta on dendrites data). These should be added and compared to the data obtained by EM.

In the data presented in Figure 6B, in mature SCs the mode of the synapse distribution is in the 45 µm bin from the soma, whereas in the immature SCs it is in the 15 µm bin. But the distributions are different in shape: in mature SCs, the distribution is quite gaussian whereas in immature SC it is skewed. It might be interesting to know the skewness of the two distributions. This is particularly evident in Figure 6A cumulative plot where the mature is very shifted to the right compared to the immature. As pointed out, interestingly the synapse distribution at the two ages (Figure 6B) is very similar to the shape of the Sholl analysis in Figure 6F. Overall how do these LM data compare to EM data quantified as suggested above?

P. 11 l. 351 "EPSCs arising from distal locations in the adult were larger and contributed relatively more to the simulated mean mEPSC waveform (Figure 7B).": This is confusing. The colour scheme (including especially the white bars) in Figure 7B is also highly confusing. Please choose a better way of presenting this key result of the paper in the text and Figure 7 – distally generated mEPSCs dominate the mean mEPSC more strongly in adult than in immature SCs, making the mean mEPSC smaller and slower in adult SCs.

P. 14 l. 467 "Voltage-clamp recordings of mEPSCs showed that their time course and amplitude are both halved during development (Figure 1).": confusing. The amplitude is halved but the rise and decay time constants are doubled.

P. 19 The authors state that "For the area measurement of synapses on soma, serial sections through three unlabeled neighbor SCs were also used to avoid potential turbulence due to the patching." (l. 627-629) Are the data presented for the PSD size of synapses onto the soma in Figure 3G from these 2 patched cells or are they from the 3 unloaded cells in Figure 4C? These should be specified and only used if the data from the patched cells in Figure 3 are statistically comparable to those from the unloaded cells in Figure 4. The data for Figure 3F and 4C have also been obtained with two different EM sectioning and imaging methods. The rationale should be explained behind passing from a 300-400 section long ribbons on tape (ATUM), that allows proper 3D reconstructions of synapses, to a "classic" serial section on grid/slots with much lower number of sections per ribbon as well as the higher risk of losing sections containing part of synapses.

*Reviewer #3 (Recommendations for the authors):*

– Modeling throughout the manuscript is done without taking into consideration the effect of series resistance. In the methods section the authors note that no series resistance compensation was used for the recordings, which considering the 6-8 MΩ tip resistance is presumably substantial. This may have a dramatic effect on the recorded current kinetics.

– Dendritic width are calculated from fluorescent images, even though the dendritic diameter is around the limit of diffraction limited techniques. The authors may have access to relevant electron microscopic data, which could be used for comparison to check for accuracy if possible.

– Dendritic impedance needs to be investigated electrophysiologically for young SCs, or the model needs to be repeated and illustrated for several different dendritic input impedance profiles.

---

## [Author Response]

Essential revisions:1. The authors should provide additional verification that the biophysical parameters of the model built for adult SCs in Abrahamsson et al., (2012) are appropriate for cells from younger animals and address the potential contribution of developmental changes in AMPAR/synaptic properties.

We thank the reviewer for pointing out that we had not clearly discussed our experimental confidence in our immature SC model parameters. The principal factors required for generating insightful predictions about dendritic integration are (a) the passive membrane properties, (b) the dendritic morphology and (c) an estimate of voltage gated channels expression.

Immature SC passive membrane properties. We estimated the passive membrane properties (membrane time constant) of immature SCs using the same method as for adult SCs (Abrahamsson et al., 2012) and found similar values. The procedure is described in the Methods. To perform model simulations considering these immature SC values we adjusted the specific membrane resistance (Rm) to match the experimentally determined membrane time constant for all biophysical models types (idealized and 3D reconstructed SC models). Developmental differences in morphology were explicitly incorporated for each model.

Immature SC dendritic tree morphology. We quantified developmental changes in dendritic morphology in two ways: (1) by directly measuring the dendritic diameter (430 measurements over 93 dendritic segments) – which critically influences dendritic filtering, and (2) by performing full dendritic tree reconstruction of two-photon images. In this manner, we were able to assess the developmental changes in dendritic tree length and branching, and synapse density. Because neither the small changes in membrane time constant or dendritic diameter could account for the developmental slowing of mEPSCs, we proposed that the larger relative number of synapses at longer electrotonic distances could account for a slower averaged mEPSC. This was confirmed by comparing model simulations from careful manual reconstructions of immature and adult morphology and synapses distribution SCs.

– We modified the Results and Discussion sections (see below for details) to reiterate the measured biophysical and morphological parameters estimated for the two developmental stages and used in numerical simulations. The principal motivation for model simulations was for illustration of proof of principle. First, we showed that immature SCs exhibit cable filtering of their synaptic responses. Secondly, we used biophysical models based on full 3D reconstructions to show that increased electrotonic filtering of mEPSCs is a result of a larger fraction of synapses at longer distance from the soma. Our conclusions are sufficiently justified by model simulations that explicitly implement the difference in synaptic conductance and morphological properties (line 533):

“However, combining experiments and simulation we showed that immature SCs have similar dendritic integration properties as adult SCs (Abrahamsson et al., 2012): (a) dendrite-evoked qEPSCs are smaller and slower than those evoked at the soma (Figure 3), consistent with cable filtering (Figure 2); (b) STP differed between dendrite and somatic stimulation and did not reveal supralinear recruitment of voltage-gated channels (Nevian et al., 2007) (Figure 8); and (c) simulated subthreshold I/Os were sublinear (Figure 8). These results are supported mechanistically by several experimental findings such as (a) the measured membrane time constant that are identical at the two developmental stages, (b) the dendritic diameters being similarly small (Figure 2), and (c) the paired-pulse facilitation difference between dendrite and somatic reproducible with a passive biophysical model like in adult SCs (Figure 8). Taken together, the data are consistent with SCs displaying little developmental changes in voltage-gated channel expression (Molineux et al., 2005) and low densities of sodium (Myoga et al., 2009) and calcium channels (Tran-Van-Minh et al., 2016).

The integration of fast synaptic events such as AMPAR-mediated EPSC is critically influenced by the intracellular resistivity, the local membrane capacitance, and dendritic diameter (Equation 2). Since internal resistivity is a challenge to estimate without dual electrode recordings, we examine a wide range of internal resistivities and found that they did not alter qualitatively our conclusions. The narrow dendritic diameters at both ages (Figure 2B) suggest that the local input resistance is only slightly different. We cannot rule out that developmental changes in gap junction expression could contribute to the maturation of SC dendritic integration, since they are thought to contribute to the axial resistivity and capacitance of neurons (Szoboszlay et al., 2016). All the recordings were made with gap junctions intact, including for membrane constant measurements. However, their expression in SCs is likely to be lower than their basket cell counterparts (Hoehne et al., 2020; Rieubland et al., 2014). Overall, the basic electrotonic machinery to filter synaptic responses is already present as soon as SC precursors reach their final location in the outer third of the molecular layer.”

Lack of voltage-gated channels in immature SCs: Dendritic nonlinearities can arise from the expression of voltage-gated channels that can amplify or counteract EPSPs. We did not explicitly examine the contribution of voltage-gated channels in immature SCs in this study, as we did for adult SCs (Abrahamsson et al., 2012). Still, for the following reasons, we did not implement voltage-gated channels in our simulations:

1) Studies on young animals (P14) and older (P24) suggest similar voltage-gated channel expression (Molineux et al. 2005), indicating that the results from Abrahamsson et al., 2012 apply to immature SC.

2) Sodium channels are expressed at low densities in SC dendrites, if at all (Myoga and Regehr, 2009).

3) Paired-pulse ratios of evoked multi-quantal EPSCs in immature SCs show no evidence of voltage-gated channel recruitment (see Nevian et al., 2007). They are also well-predicted by our passive biophysical models, which we have now added to Figure 8 (panel B).

4) Most of the study focuses on quantal EPSCs, which are less likely to depolarize dendrites to recruit voltage-gated channels.

Regarding the potential contributions of developmental changes in AMPAR to EPSC. At some synapses, properties of synaptic AMPAR have been shown to change during development, underlying a developmental speeding of mEPSCs. EPSC in immature SC have been shown to be mediated by CP-AMPAR predominantly containing GluR3 associated with TARP γ2 (Stargazin) and TARP γ7 (Soto et al., 2007; Bats et al., 2012). During development, GluR2 subunits are inserted into synaptic AMPARs in an activity-dependent manner (Liu et al., 2000), affecting calcium permeability of the receptors (Liu et al., 2002: Soto et al. 2007) without impacting EPSC kinetics (Liu et al., 2002) or channel conductance (Soto et al. 2007). Consistent with these findings, we did not observe any changes qEPSC kinetics for somatic synapses between immature and adult SCs. We, therefore, concluded that the developmental slowing of mEPSCs is unlikely due to alterations in AMPAR subunits or associated proteins.

We have modified the Results.

“This reduction in synaptic conductance could be due to a reduction in the number of synaptic AMPARs activated and/or a developmental change in AMPAR subunits. SC synaptic AMPARs are composed of GluA2 and GluA3 subunits associated with TARP γ2 and γ7 (Bats et al., 2012; Liu and Cull-Candy, 2000; Soto et al., 2007; Yamazaki et al., 2015). During development, GluR2 subunits are inserted to the synaptic AMPAR in an activity-dependent manner (Liu and Cull-Candy, 2002), affecting receptors calcium permeability (Liu and Cull-Candy, 2000). However, those developmental changes have little impact on AMPAR conductance (Soto et al., 2007), nor do they appear to affect EPSC kinetics (Liu and Cull-Candy, 2002); the latter is consistent with our findings. Therefore the developmental reduction in postsynaptic strength most likely results from fewer AMPARs activated by the release of glutamate from the fusion of a single vesicle.”

We have also modified the Discussion.

“Comparison of synaptic properties between immature and adult SCs

Whole-cell voltage-clamp recordings in immature and adult SCs showed a developmental decrease in mEPSC amplitude and slowed kinetics (Figure 1). These results contrast with findings in other neurons that show faster mEPSCs during maturation due to changes of AMPAR subunits, vesicular content of neurotransmitter, and/or synapse structure (Cathala et al., 2005; Chen and Regehr, 2000; Koike-Tani et al., 2005; Yamashita et al., 2003). Knowing that dendritic inputs could be electrotonically filtered, we took advantage of the ability to selectively stimulate somatic synapses to isolate somatic qEPSCs for comparison between the two ages. Evoked EPSCs showed a developmental reduction in amplitude (~ 40%), with no change in kinetics (Figure ). This likely results from the smaller adult PSD size (Figure 4), and hence a lower AMPAR number (Masugi-Tokita et al., 2007), rather than a developmental reduction in the peak glutamate concentration at the postsynaptic AMPARs (Figure 1) or change in AMPAR subunits composition (see Results section).”

2. Broaden the scope of the conclusions by exploring additional implications of the observed developmental changes, for example, by testing whether inclusion of NMDARs alters the relative strength of distal inputs, how gap junction coupling between SCs would affect the results, and testing different dendritic impedance scenarios.

NMDARs:

In this manuscript, we focused on the integration of single-vesicle release events, which comprise those synaptic events during low-frequency stimulation (see below for a detailed justification). NMDARs are expressed in SC, but due to extrasynaptic localization, are not significantly activated by single vesicle release of neurotransmitter. This observation has been well-documented by several labs in immature (Glitsch et al., 1999; Carter et al., 2000; Clark et al., 2002) and adult SC (Abrahamsson et al., 2012, Tran-Van-Minh et al., 2016), which have shown that EPSC and EPSP integration is insensitive to NMDAR antagonist. As a result, NMDARs would not alter the relative strength of distal inputs under low-frequency stimulation and thus will not contribute to synaptic integration of quantal EPSPs in either somata or dendrites.

It is now addressed in the Result:

“We did not examine NMDAR currents since they are located extrasynaptically and do not contribute to postsynaptic current under low intensity and low-frequency stimulation (Carter and Regehr, 2000; B. A. Clark and Cull-Candy, 2002; Tran-Van-Minh et al., 2016). Therefore, AMPAR-mediated mEPSCs can provide an unbiased assessment of the effective distribution of postsynaptic strengths throughout the entire somato-dendritic compartment.”

Gap junctions:

All recordings were performed without perturbing gap junction and therefore implicitly account their contribution to the passive properties of SCs. Gap junctions are expected to contribute to the axial resistivity and capacitance of neurons (Szoboszlay et al., 2016). Unlike in Golgi cells (Szoboszlay et al., 2016), the precise location and conductance of gap junctions in SCs is not known and thus difficult to implement accurately in biophysical models. Moreover, Gap junctions in SCs are likely to be much less prominent than their basket cell and Golgi cell counterparts both in both immature (Rieubland 2015) and mature SCs (Hoehne et al., 2020). We, therefore, think that accurate and explicit modeling of gap junctions in SCs is beyond the scope of this study. Nevertheless, because we matched the membrane time constant of the model to that experimentally measured with any gap junctions intact, we do not expect any impact on our conclusions.

We now discuss the implication of gap junctions in the Discussion:

“We cannot rule out that developmental changes in gap junction expression could contribute to the maturation of SC dendritic integration, since they are thought to contribute to the axial resistivity and capacitance of neurons (Szoboszlay et al., 2016). All the recordings were made with gap junctions intact, including for membrane constant measurements. However, their expression in SCs is likely to be lower than their basket cell counterparts (Hoehne et al., 2020; Rieubland et al., 2014).”

Testing dendritic impedance scenarios:

The most important parameter required for accurate estimation of the local impedance is the dendritic diameter. For fast AMPAR mediated synaptic conductances, cable filtering primarily depends on the internal resistivity, the dendritic diameter, and the membrane capacitance (Abrahamsson et al., 2012 and equation 2). Lack of influence of Rm was verified in adult SC models (see the panel A of Figure S4 in Abrahamson et al. 2012). For this reason, we did not think it was necessary to explore this issue here. The specific membrane capacitance is assumed to be 0.9 uF/um2, a well-accepted value for rodent neurons. In simulations, we varied Ri over a range of published estimates, which we show as errors on most simulation plots. Because variations in Ri do not alter our conclusions, the critical parameter to be estimated is the dendritic tree morphology, particularly the dendritic diameter, which critically influences input impedance. We, therefore, conscientiously measured dendritic diameter in young SCs using confocal microscopy, the same method used for adult dendrites (Abrahamsson et al., 2012).

We have added the following sentence to the Discussion section:

“The integration of fast synaptic events such as AMPAR-mediated EPSC is critically influenced by the intracellular resistivity, the local membrane capacitance, and dendritic diameter (Equation 2). Since internal resistivity is a challenge to estimate without dual electrode recordings, we examine a wide range of internal resistivities and found that they did not alter qualitatively our conclusions. The narrow dendritic diameters at both ages (Figure 2B) suggest that the local input resistance is only slightly different.”

Equation 2 from manuscript:λAC≈d4πfRiCm

3). Provide all the technical details that would allow reproduction of the experiments. More methodological information for both the electrophysiology and simulations need to be included, including series resistance errors.

Methodological information requested by reviewers for both the electrophysiology and simulations have been added to the method section.

Reviewer #1 (Recommendations for the authors):1) Line 150-158. Foster et al., 2005 shouldn't be used to support exclusive uni-vesicular release at PF-MLI synapses, since it addresses multi-vesicular release and AMPAR saturation at PF-PC synapses.

Indeed, our sentence was misleading. We reworded it to explain that even though GC-SC synapses exhibit multivesicular release, the low release probability of GC synapses implies that, on average, only one vesicle is released per synaptic connection per action potential (AP). We, therefore, postulate that the study of dendritic integration of quantal EPSCs is physiologically relevant. We corrected the sentence in the manuscript:

Line 171:

“Because of their low release probability, excitatory synapses formed by granule cell axons (parallel fibers, PFs) release on average only one synaptic vesicle per synaptic contact, despite the presence of multiple release sites per synaptic contact (Foster et al., 2005).”

In general, a central assumption that mEPSCs provide an unbiased assessment of the effective synaptic strength is not convincing because there is abundant evidence that PF-MLI synaptic contacts are heterogeneous in terms of the number of vesicles released per action potential (Malagon et al., 2016; Pulido and Marty, 2019; Vaden et al., 2019).

We thank the reviewer for pointing out these studies. The studies from Dr. Marty’s group used 3 mM extracellular calcium, which would un-physiologically increase the fraction of docked vesicles released at single contacts and the vesicle fusion probability, thereby increasing quantal content for each AP. The physiological extracellular calcium concentration is thought to be between 1.2 and 1.5 mM. Finally, the Pulido study was performed on GABAergic synapses formed between molecular interneurons. The Vaden et al., 2019 study of climbing fiber to Purkinje cells synapses, neither of which is comparable to our study.

In summary, we prefer to comment on the physiological relevant quantal content described in Foster et al., 2005, which uses 1.2 mM extracellular calcium.

Nahir and Jahr (2013) provide support that a majority of evoked release is mediated by a single vesicle at PF-MLI synapses in adolescent rats, but considering the goals of this manuscript to assess the contribution of 'synaptic strength" over development, the authors should address whether this variable contributes to the differences between mEPSC and qEPSCs as well as qEPSCs over development.

We do not understand the reference to Nahir and Jahr (2013) with respect to single vesicle release. Nahir and Jahr demonstrated both multivesicular and univesicular release. However, importantly, when the number of vesicles released per AP per synapse increases, partly due to presynaptic facilitation of release and temporal summation of residual glutamate, spillover onto extrasynaptic receptors is observed. They and others (Clark et al., 2002) have shown that this does not occur in response to single vesicle fusion. Since in this study, mEPSCs and qEPSCs are responses to the release of single vesicles, we focused on the influence of dendritic morphology contributed to the filtering of single vesicle release events without considering NMDAR activation. Consideration of the nonlinear recruitment of NMDARs during high-frequency burst stimulation is beyond the scope of this study. Nevertheless, the slow time course of NMDA receptor-mediated synaptic currents would produce synaptic responses that are less sensitive to cable filtering in SCs.

We have modified the Results section accordingly:

“We did not examine NMDAR currents since they are located extrasynaptically and do not contribute to postsynaptic current under low intensity and low-frequency stimulation (Carter and Regehr, 2000; B. A. Clark and Cull-Candy, 2002; Tran-Van-Minh et al., 2016)”.

In our study, we define mEPSCs as spontaneous synaptic currents that arise from synapses throughout the dendritic tree in the absence of presynaptic APs (recordings performed in the presence of TTX), whereas qEPSCs are elicited from specific locations along the somatodendritic axis following extracellular stimulation of a presynaptic AP, but under very low release probability conditions such that only single vesicle release events are evoked. Thus, both synaptic current types can be used to assess postsynaptic strength. The contribution of release probability and the number of release sites are not pertinent to our study of mEPSCs and qEPSCs.

We have also clarified throughout the manuscript that our study primarily examines

developmental alterations in postsynaptic strength.

2) Figure 2. It looks like the amplitude of the simulated EPSC drops off faster as a function of distance in the mature vs. immature SC (these data should be presented on the same scale). Is this due to increased branching in the mature cell that in turn mediates the difference in Figure 8D? Please discuss more explicitly.

We replotted the curves on the same scale and there is no notable difference (see Figure 2).

3) Figure 7. It would strengthen the conclusion to see this model validated experimentally with electrical stimulation, such as by plotting the qEPSC amplitudes and kinetics as a function of distance of the stim electrode from the soma. It seems like these data might already exist, but aren't presented directly.

A systemic study of the distance-dependence of quantal EPSCs was not performed. Such an experiment requires two-photon imaging for precise estimates of stimulus electrode to soma distance and is not available in the PI’s laboratory. Nevertheless, dendritic stimulation was performed by placing the stimulation electrode on the outer third of the dendritic field on isolated dendrites. However, in a subset of recording combined with 2P imaging, we obtained dendritic qEPSC with a peak amplitude of 42.5 +/- 7 pA , a 0.24 +/- 0.02 ms RT and 0.83 +/- 0.07 ms halfwidth at a distance of 51.57 μm (range 43 to 63 um, n = 5), similar to the data set described in the manuscript. Since those data represent only represent a subset of results, they were not included in the manuscript. We, however, clarified in the manuscript the location of the dendritic simulation:

“In contrast, stimulation of distal dendritic synapses, in the outer third of the dendritic field, produced somatically recorded qEPSCs that were significantly smaller (mean amplitude: 46 ± 3 pA) and slower (10-90 % rise time of 0.24 ± 0.012 ms and a half-width of 0.88 ± 0.04 ms; n = 18; all P<0.05, Figure 3C, 3D). Simulated qEPSCs generated from a dendirtic location 45 μm from the soma (Figure 2C) exhibit amplitude and kinetics values with the range of experimental values. Taken together, these results are consistent with cable filtering of EPSCs as they propagate along dendrites in immature SCs.”

Specific validation of the model by direct comparison of simulations and recorded EPSCs is a standard challenge in studying dendritic integration. The variability of passive properties combined with morphological difference across dendrites and cells and variability in synaptic strength make it difficult to predict specific experimental results from simulations using mean parameters obtained from a different population of neurons. While these challenges are not unsurmountable, they are well beyond the scope of this manuscript. However, simulations by immature SC models reproduce the reduction in qEPSCs amplitude and slowing in their kinetics that falls within the range (+/- SD) of the experimental qEPSCs evoked at distal dendritic and somatic synapses. In addition, the simulations in Figure 7 using the reconstructed morphology of a single immature SC and a single adult SC reproduce experimental mEPSC data. Nevertheless, to better describe how the simulation match experiments, we modified Figure 2C by adding plots illustrating the distance-dependence of EPSC amplitude, rise time, and half-width and now directly compare measured qEPSCs to those predictions:

“Simulated qEPSCs were large and fast for synapses located at the soma (magenta trace, Figure 2C), but qEPSCs evoked from synapses located on the dendrites (grey trace; at 45 μm from the soma, Ri of 150 Ohm·cm) were 48% smaller and showed a 195% slower rise time and a 180% slower half-width. This dendritic filtering was also associated with a large increase in the local synaptic depolarization (gren trace, Figure 2C) that would substantially reduce the local driving force during synaptic transmission onto dendrites, potentially causing a sublinear read-out of the underlying conductance, as observed in adult SCs (Abrahamsson et al., 2012; Tran-Van-Minh et al., 2016). Examination of the amplitude, rise time, and half-width as a function of synapse distance shows that beyond ~40 μm there is little additional cable filtering (Figure 2C, right).”

“Similarly, activation of synaptic inputs along a dendrite at increasing distance from the soma produced soma-recorded qEPSCs that become smaller and slower with distance (Figure 2F), similar to those in the idealized passive model (Figure 2C) with a dendritic diameter matching that obtained from confocal images (Figure 2B). We also simulated qEPSCs from a reconstructed adult SC (Figure 2G, 2H), which showed a similar distance-dependent decrease in amplitude as for the immature SC (Figure 2F). These simulations suggest that, like their adult counterparts, the passive morphometric characteristics of immature SCs should also produce significant cable filtering of both the amplitude and time course of EPSCs.”

“In contrast, stimulation of distal dendritic synapses, in the outer third of the dendritic field, produced somatically recorded qEPSCs that were significantly smaller (mean amplitude: 46 ± 3 pA) and slower (10-90 % rise time of 0.24 ± 0.012 ms and a half-width of 0.88 ± 0.04 ms; n = 18; all P<0.05, Figure 3C, 3D). Simulated qEPSCs generated from a dendirtic location 45 μm from the soma (Figure 2C) exhibit amplitude and kinetics values with the range of experimental values. Taken together, these results are consistent with cable filtering of EPSCs as they propagate along dendrites in immature SCs.”

Including experimental data like what is shown in Figure 1 and 2 of Abrahammson et al., but in the adolescent SCs would validate that the previously established model still applies in cells from younger animals, and help rule out other contributions from AMPARs or dendritic membrane properties, or synaptic glutamate concentration (see pt 1).

Regarding a possible developmental change in synaptic glutamate concentration, we ruled this out directly by performing low-affinity antagonist experiments. Postsynaptic AMPAR blockade using γDGG produced an identical reduction in EPSC peak amplitude in immature and adult SCs (Figur 1E), with no effect on kinetics (Figure 1F). In addition, we found that the ~30% in qEPSC amplitude observed at the soma (Figure 4B) is associated with a similar reduction in PSD size (figure 4C) indicative of a smaller number of AMPAR activated upon glutamate release rather than a change in the glutamate concentration waveform. This point is discussed in the Results.

We also ruled out a contribution of different AMPAR subunits because qEPSCs recorded at the soma have identical kinetics at both ages (Figure 4B). It, therefore, seems unlikely that development changes in AMPAR subunits contribute to the observed kinetic changes, consistent with the observation that the insertion of GluR2 subunits to the synaptic AMPAR does not affect EPSC kinetics (Liu et al., 2002). This point is now discussed in the Results.

We ruled out developmental changes in passive membrane properties by examining membrane time constant, which we found almost identical for the two developmental stages. See the response to the Editors for a more detailed reply.

4) Figure 8 B,C,D. It is not surprising that the same biophysical model with smaller dendritic branches recapitulates the behavior of the adult SC with minor quantitative differences (it is hard to imagine a different outcome). But one wonders whether this simulation can be verified experimentally

Indeed, it is possible to perform two-photon uncaging to examine the altered sublinearity between immature and adult SCs, but this technique is not available currently in the PI’s laboratory. We agree with the reviewer that experimental validation is important. We, therefore, demonstrated that dendritic the EPSC paire-pulse ratio is lower than at the soma due to sublinear summation (Figure 8A), as shown in adult SCs (Abrahamsson et al., 2012). Moreover, we now show in a new panel (Figure 8B) that the dendritic PPR from immature SCs can be reproduced with our passive model of immature SCs. Description of the new panel starts on line 441.

or perhaps more importantly, whether conclusions would be generalizable if NMDAR activation (which seems likely to change over maturation) is considered. The exclusive focus on AMPAR-only integration seems of limited relevance for understanding the computational changes occurring over SC development.

We think that our study of exclusively AMPAR mediated synaptic responses is relevant since NMDAR are extrasynaptic and require long, high-frequency trains of presynaptic stimuli for their activation. This observation has been well documented by several labs in immature (Glitsch et al., 1999; Carter et al., 2000; Clark et al., 2002) and adult SCs (Abrahasson et al., 2012; Tran-Van-Minh et al., 2016). These studies also showed that EPSCs evoked by single and paired extracellular stimulation are insensitive to NMDAR antagonists. As a result, NMDAR would not alter the relative strength of distal inputs and could not contribute to synaptic integration on both somatic and dendritic EPSP. In summary, NMDAR activation is not relevant for our study of developmental determinants of changes in the strength of postsynaptic quantal responses or EPSCs following single or pairs of presynaptic stimuli. Understanding the contribution of NMDARs to synaptic strength during train stimuli is titillating but beyond the scope of this study.

Nevertheless, we have added text to the Discussion justify our emphasis on AMPAR-mediated EPSCs:

“We did not examine NMDAR currents since they are located extrasynaptically and do not contribute to postsynaptic current under low intensity and low-frequency stimulation (Carter and Regehr, 2000; B. A. Clark and Cull-Candy, 2002; Tran-Van-Minh et al., 2016)”

Reviewer #2 (Recommendations for the authors):P. 7 l. 197 the different sections of the dendrites per cell should have been quantified separately to determine whether there are variations among dendrites in the same cell and among different cells. The question of how locations for diameter measurements (see also the next point) were chosen is relevant in particular when adult SCs presenting higher dendritic complexity reflected in more branches are being compared to immature SCs. It would be worth considering to break down the diameters per branch order, instead of collating all data in a global mean (see below P15, l. 429-430). All this could affect the properties of dendrites as well as the distribution of synapses along the dendrites (see Figure 2B p33).

We thank the reviewer for this comment and have analyzed the dendritic widths according to location along the dendrite. We have added 2 new panels to Figure 2B: (a) one describing the variability in diameter observed between dendritic segments that reflects the variability between dendrites in the same cell and between different SC, and (b) a second describing dendritic width as a function branch order showing a small reduction of dendritic segments as a function of distance from the soma (0.35 to 0.6 μm). This variation of diameter would not alter cable filtering properties significantly since dendritic filtering is comparable for diameters ranging from 0.3 to 0.6 μm, (Figure 4G and 4H, Abrahamsson et al., 2012).

P. 31 Figure 2A,B how did the authors ensure fair sampling of the dendritic tree for the measurement of dendritic diameters?

We arbitrarily selected pixel intensity line profiles along the dendritic tree (n=430) and average the profiles from the same dendritic branch segment (n=93) from 18 immature SCs. We now describe this procedure more clearly in the Methods:

“Dendritic diameters were estimated from the full-width at half maximum (FWHM) of pixel intensity line profiles on 1 μm segments of dendrites, made perpendicular to dendritic length using the same procedure as for adult SCs (Abrahamsson et al., 2012). In brief, we selected dendritic segments for measurement throughout the dendritic tree to homogeneously sample the diversity of diameters. We collected 430 line profiles from 18 immature SCs, then average the profiles from the same dendritic segments (n=93). The imaging resolution within the molecular layer was estimated from the width of intensity line profiles of SC axons. The FWHM was 0.30 +/- 0.01 μm (n = 57 measurements over 16 axons) and a mean of 0.27 +/- 0.01 μm (n = 16) when taking into account the thinnest section for each axon. Only 2% of all dendritic measurements are less than 270 nm, suggesting that the dendritic diameter estimation is hardly affected by the resolution of our microscope.”

P. 31 Figure 2B Compare to Abrahamsson et al., 2012 Figure 4B: it would be helpful to see the two distributions (immature and mature SC) together.

We have added the distribution from Abrahamsson et al., to Figure 2B.

P. 9 l. 280 "PSD area was constant…." But in Figure 3G there is no quantification of this. It is just an observation. Provide quantitative data about PSD are at the soma vs dendrites and distance from the soma, as well as cumulative plots and bin distributions instead of Figure 3G (as done for LM data in Figure 6).

The reviewer makes a good point; the linear correlation estimate was inadvertently omitted from Figure 3G. We elected not to bin the data in order to correctly weigh the analysis according to the number of sample measures at each distance. The linear fit was added to Figure 3G and the following text added to the figure legend: “The green line is a linear fit through all the points (R2 = 0.001, p = 0.85).”

P. 10 l. 298-299 n = 83 from 3 cells (see Figure 4C legend) vs n = 97 from 2 cells from Abrahamsson et al., 2012. N = 83 from 3 cells from immature when n = 97 from 2 cells in mature: is the number of excitatory synapses onto the soma the same in immature vs mature SC? A density measurement would be more appropriate to compare variations, if any.

Because not all the synapses on the soma were reconstructed in the EM samples, the number of somatic synapses is best estimated using the light microscopy approach. The N=83 from 3 cells is just the number of sampled synapses and not all synapses on soma. Therefore, we cannot compare it with n = 97 from 2 cells from Abrahamsson et al.

P. 9 l. 277: Why did the authors not compare (and pool if statistically possible) the data from the 3 unloaded SCs (Figure 4C) and the 2 loaded cells (used in Figure 3F)?

Because of different methods used for the two groups of cells (classical serial sections observed with TEM for the 3 unloaded and ATUMtome sections observed with SEM for the 2 loaded cells), it is not appropriate to pool the data sets. Between the three unloaded SCs, there is no significant difference in PSD size. For the two loaded SCs, there is also no significant difference in PSD size (p = 0.19). However, the data of PSD sizes in the 3 unloaded and 2 loaded SCs are significantly different (0.039 ± 0.002 μm^2^, n = 83 vs. 0.020 ± 0.001 μm^2^, n = 137, p<0.0001), perhaps due to the different methods.

We have added this information to the methods:

“In order to compare synapse area measurement comparable to those done on adult soma (Abrahamsson et al., 2012), we performed serial sections and then standard transmission electron microscopy (TEM) on three unlabeled SCs (which avoids potential distortions due to the patching).”

P. 10 l. 306-311 "In addition to cable filtering of synaptic currents, a neuron's somatic response to dendritic input is also affected by synapse distribution within its dendritic tree,….." and P. 10 l. 320-321 "To test this hypothesis, we examined the distribution of excitatory synaptic inputs along the somato-dendritic compartment in immature and adult SCs." Light microscopy was utilized to quantify these parameters (Figure 6). These parameters should also be quantifiable from the EM data the authors already have. From these data it should be possible to quantify and plot "synapse number vs distance from the soma". These data should then be compared to the data presented at P. 11 l. 326-336, as well as to the previous EM data about PSD size once these have also been quantified (P. 9 l. 280). EM data should accompany the LM data even more so when the authors themselves point out in Methods (l. 719-722) "This approach is limited by the resolution of 2P fluorescence imaging to differentiate individual synapses within clusters and thus may result in an underestimate of absolute synapse density, but allowed for an unbiased estimate of synapse distributions at the two developmental stages."

Unfortunately, we could not use the EM data appropriately for “synapse number vs distance from the soma” because not all synapses along the dendritic tree were sampled. We used 300-400 serial sections obtained by ATUMtome as described in the Methods section but the number of sections was not sufficient for a complete reconstruction of soma and dendrites of single cells. We used the LM method to overcome this limitation for a systematic and unbiased comparison of synapse (puncta) densities between immature and adult SCs.

P. 11 l. 323-324 "We then mapped Venus-tagged PSD95 puncta associated with the somata and dendritic trees of Alexa 594-filled immature and adult SCs (Figure 5A-D)" and l. 326-328 "Venus-tagged PSD95 puncta within the soma and dendritic tree of 9 immature and 8 adult 3D-reconstructed SCs (Figure 5G), showed ~ 80 % more puncta on adult SCs (582 {plus minus} 48 puncta vs. 324 {plus minus} 21 in immature SC, p<0.05)."No data about the PSD95 puncta on somata only are presented (i.e. separated from the puncta on dendrites data). These should be added and compared to the data obtained by EM.

The number of somatic synapses detected using LM is now presented in panel B of Figure 6 and listed in the data source file. We found 27 +/- 3.2 synapses in immature SC versus 21.17 +/- 2.5 synapses in Adult SC. These values have been added to the text (Line 375):

“Venus-tagged PSD95 puncta within the soma and dendritic tree of 9 immature and 8 adult 3D-reconstructed SCs (Figure 5G), showed ~ 80 % more puncta on adult SCs (582 ± 48 puncta vs. 324 ± 21 in immature SC, p<0.05; of which 27 ± 3.2 and 21.17 ± 2.5 are at the soma of immature and adult SC, respectively).”

Because not all the synapses on the soma were sampled within the serial sections, we considered LM our only reliable somatic synapse number estimate.

In the data presented in Figure 6B, in mature SCs the mode of the synapse distribution is in the 45 µm bin from the soma, whereas in the immature SCs it is in the 15 µm bin. But the distributions are different in shape: in mature SCs, the distribution is quite gaussian whereas in immature SC it is skewed. It might be interesting to know the skewness of the two distributions. This is particularly evident in Figure 6A cumulative plot where the mature is very shifted to the right compared to the immature. As pointed out, interestingly the synapse distribution at the two ages (Figure 6B) is very similar to the shape of the Sholl analysis in Figure 6F. Overall how do these LM data compare to EM data quantified as suggested above?

We assessed the skewness of the distribution of PDS95 puncta and found that there is a small difference between adult (basically no skew) and the immature (slight positive skew). The values are added to the Figure legend. We also compared the skewness of the PSD95 distributions to the skewness of the dendritic segment distributions for the two ages and we found comparable skewness supporting further our finding that puncta density remained constant across the dendritic tree at both ages.

Pearson’s median skewness (3 (mean − median)/standard deviation):

Immature Adult

PSD95 dendrites PSD95 dendrites

0,38 0,32 -0,10 -0,08

We thank the reviewer for his suggestion and have added a sentence to point out the similarity of skew in the dendritic segment and PSD-95 labeled puncta distributions (see Results):

“The similarity between the shapes of synapse (Figure 6B) and dentric segment (Figure 6C) distributions was captured by a similarity in their skewness (0.38 vs. 0.32 for both distributions in immature and -0.10 and -0.08 for adult distributions)”

As stated above we could not appropriately use the EM data for “synapse number vs distance from the soma” because not all synapses were sampled by the serial sections.

P. 11 l. 351 "EPSCs arising from distal locations in the adult were larger and contributed relatively more to the simulated mean mEPSC waveform (Figure 7B).": This is confusing. The colour scheme (including especially the white bars) in Figure 7B is also highly confusing. Please choose a better way of presenting this key result of the paper in the text and Figure 7 – distally generated mEPSCs dominate the mean mEPSC more strongly in adult than in immature SCs, making the mean mEPSC smaller and slower in adult SCs.

The text has been modified in the result section:

“Assuming that mEPSCs are generated randomly with an equal probability at all synapses, we generated a simulated mean mEPSC by summing the qEPSCs generated at each distance (qEPSC_d_)each weighted by its relative frequency according to the synapse distribution (Figure 7B, right panel). For each dendritic segment, the obtained weighted qEPSC_d_ describes its relative contribution to the mean mEPSC waveform (Figure 7B, left panel). One can see that qEPSC_d_s arising from distal locations in the adult were relatively larger in amplitude as compared to immature SCs and therefore would contributed more to the simulated mean mEPSC waveform. As an example, the contribution of the weighted qEPSC_d_ at 45 μm to mEPSC was small in immature SC while it was more prominent in adult SC (compare the green traces in Figure 7B). As a result, the mean mEPSC was smaller in adult SCs than in immature SCs (26.0 ± 0.6 pA, n = 9 dendrites vs. 52.2 ± 0.4 pA, n = 7 for R_i_ 150) and its time-course had slower rise (10-90 % rise time = 0.24 ± 0.05 vs. 0.17 ± 0.01 ms) and decay (half-width = 1.11 ± 0.04 vs. 0.77 ± 0.01 ms; for all P<0.05). Thus, distally generated qEPSC_d_s dominate the mean mEPSC more strongly in adult than in immature SCs, making the mean mEPSC smaller and slower in adult SCs.”

The color scheme of the Figure 7B has been explained in the Figure legend:

“For clarity only a subset of normalized qEPSC_d_ are displayed with a different color for somatic and each 10 μm dendritic segments. Hollow bars in the histogram correspond to distance not illustrated with a normalized qEPSC_d_ on the left panel”

P. 14 l. 467 "Voltage-clamp recordings of mEPSCs showed that their time course and amplitude are both halved during development (Figure 1).": confusing. The amplitude is halved but the rise and decay time constants are doubled.

This sentence has been modified:

“Whole-cell voltage-clamp recordings in immature and adult SCs showed a developmental descrease in mEPSC amplitude and slowing of their kinetics (Figure 1).”

P. 19 The authors state that "For the area measurement of synapses on soma, serial sections through three unlabeled neighbor SCs were also used to avoid potential turbulence due to the patching." (l. 627-629) Are the data presented for the PSD size of synapses onto the soma in Figure 3G from these 2 patched cells or are they from the 3 unloaded cells in Figure 4C? These should be specified and only used if the data from the patched cells in Figure 3 are statistically comparable to those from the unloaded cells in Figure 4. The data for Figure 3F and 4C have also been obtained with two different EM sectioning and imaging methods. The rationale should be explained behind passing from a 300-400 section long ribbons on tape (ATUM), that allows proper 3D reconstructions of synapses, to a "classic" serial section on grid/slots with much lower number of sections per ribbon as well as the higher risk of losing sections containing part of synapses.

We are sorry about the confusion. The data of the PSD size in Figure 3G are from the 2 loaded cells, using“classical”serial sectioning and TEM imaging methods, as was performed for Abrahamsson et al., 2012. This was essential for the comparison of PSD size between immature and adult SCs. Unfortunately, the data obtained with ATUMtome and SEM were statistically different from those of the unloaded cells, as mentioned above, and could not be pooled together.

Reviewer #3 (Recommendations for the authors):– Modeling throughout the manuscript is done without taking into consideration the effect of series resistance. In the methods section the authors note that no series resistance compensation was used for the recordings, which considering the 6-8 MΩ tip resistance is presumably substantial. This may have a dramatic effect on the recorded current kinetics.

Simulations with an idealized SC model were performed using an Rs of 20 Mohm, whereas modeling with the reconstructed SC were performed using a series resistance of 16 Mohm. The mean experimental Rs was 16 Mohm.

– Dendritic width are calculated from fluorescent images, even though the dendritic diameter is around the limit of diffraction limited techniques. The authors may have access to relevant electron microscopic data, which could be used for comparison to check for accuracy if possible.

Dendritic width was measured using the higher resolution technique of single photon confocal microscopy. Since SC axon diameters are less than 200 nm, we used them to test the resolution of our microscope within the same cerebellar slice and region. The FWHM of line intensity profiles was 0.295 +/- 0.009 μm (n = 57 measurements over 16 axons) with a mean of 0.27 +/- 0.007 μm (n = 16) when taking into account the thinnest section for each axon. Those measures are smaller 98% of the FWHMs from SC dendrites and shows that our dendritic diameter estimation is not affected by the resolution of our microscope. We therefore assumed that the larger FWHMs of line intensity profiles measured from dendrites is an accurate measure of their width.

We clarified this procedure in the Methods:

“Dendritic diameters were estimated from the full-width at half maximum (FWHM) of pixel intensity line profiles on 1 μm segments of dendrites, made perpendicular to dendritic length using the same procedure as for adult SCs (Abrahamsson et al., 2012). In brief, we selected dendritic segments for measurement throughout the dendritic tree to homogeneously sample the diversity of diameters. We collected 430 line profiles from 18 immature SCs, then average the profiles from the same dendritic segments (n=93). The imaging resolution within the molecular layer was estimated from the width of intensity line profiles of SC axons. The FWHM was 0.30 +/- 0.01 μm (n = 57 measurements over 16 axons) and a mean of 0.27 +/- 0.01 μm (n = 16) when taking into account the thinnest section for each axon. Only 2% of all dendritic measurements are less than 270 nm, suggesting that the dendritic diameter estimation is hardly affected by the resolution of our microscope.”

EM diameter estimates are likely to be in error due to fixation artifacts that generally result in tissue shrinkage. The use of confocal microscopy also allowed us to compare immature and adult SC using the same method and to minimize experimental biases between the two measurements.

– Dendritic impedance needs to be investigated electrophysiologically for young SCs, or the model needs to be repeated and illustrated for several different dendritic input impedance profiles.

As indicated above we estimated several important electrophysiological and morphological parameters in SCs, including membrane time constant, synaptic conductance amplitude and time course, dendrite diameter (and now the diameters per branch order and inter- and intra-neuronal diversity – see added panels to Figure 2B and our response to reviewer 2), length and branching, which when incorporated into biophysical models reproduced the slowing of mEPSCs dendrite evoked quantal EPSCs, and paired-pulse facilitation within dendrites. All of the simulations were performed with multiple internal resistivities, as this could not be measured. Variations across values in the literature of this parameter did not alter our conclusions. As mentioned in responses to Reviewer 1, we do not think it is possible to perform additional experiments to constrain parameters due to the natural variability of dendritic morphology and AMPAR content at each synapse. Moreover, our goal was to use model simulations to support our conclusion that dendritic branching has a surprising effect on dendritic computations due to the larger number of distal synapses without changes in cable properties.

Thus, the dendritic input resistance is best examined using simulations that are grounded by experimentally validated parameters. The dendritic diameter is one of the most important and one we experimentally determined.